# Photochemical Evolution of the 2013 California Rim Fire: Synergistic Impacts of Reactive Hydrocarbons and Enhanced Oxidants

Glenn M. Wolfe[1], Thomas F. Hanisco[1], Heather L. Arkinson[2], Donald R. Blake[3], Armin Wisthaler[4,5], Tomas Mikoviny[5], Thomas B. Ryerson[6,7,*], Ilana Pollack[7,**], Jeff Peischl[7], Paul O. Wennberg[8,9], John D. Crounse[8], Jason M. St. Clair[8,***], Alex Teng[8,****], L. Gregory Huey[10], Xiaoxi Liu[10,*****], Alan Fried[11], Petter Weibring[11], Dirk Richter[11], James Walega[11], Samuel R. Hall[12], Kirk Ullmann[12], Jose L. Jimenez[7,13], Pedro Campuzano-Jost[7,13], T. Paul Bui[14], Glenn Diskin[15], James R. Podolske[14], Glen Sachse[15,16,†], and Ronald C. Cohen[17,18]

[1]Atmospheric Chemistry and Dynamics Laboratory, NASA Goddard Space Flight Center, Greenbelt, MD, USA
[2]Department of Oceanic and Atmospheric Science, University of Maryland, College Park, MD, USA
[3]Department of Chemistry, University of California Irvine, Irvine, CA, USA
[4]Institute for Ion Physics and Applied Physics, University of Innsbruck, Innsbruck, Austria
[5]Department of Chemistry, University of Oslo, Oslo, Norway
[6]Chemical Sciences Laboratory, NOAA, Boulder, CO, USA
[7]Cooperative Institute for Research in Environmental Sciences, University of Colorado Boulder, Boulder, CO, USA
[8]Division of Geological and Planetary Sciences, California Institute of Technology, Pasadena, CA, USA
[9]Division of Engineering and Applied Science, California Institute of Technology, Pasadena, CA, USA
[10]School of Earth and Atmospheric Sciences, Georgia Institute of Technology, Atlanta, GA, USA
[11]Institute of Arctic and Alpine Research, University of Colorado, Boulder, CO, USA
[12]Atmospheric Chemistry Observations and Modeling Laboratory, National Center for Atmospheric Research, Boulder, CO, USA
[13]Department of Chemistry, University of Colorado Boulder, Boulder, CO, USA
[14]Atmsopheric Sciences Branch, NASA Ames Research Center, Moffett Field, CA, USA
[15]NASA Langley Research Center, Hampton, VA, USA
[16]National Institute of Aerospace, Hampton, VA, USA
[17]Department of Earth and Planetary Sciences, University of California, Berkeley, CA, USA
[18]College of Chemistry, University of California, Berkeley, CA, USA
*Now at Scientific Aviation, Boulder, CO, USA
**Now at Department of Atmospheric Science, Colorado State University, Fort Collins, CO, USA
***Now at Joint Center for Earth Systems Technology, University of Maryland Baltimore County, Baltimore, MD, USA
****Now at Fifty Years, San Francisco, CA, USA
*****Now at California Air Resource Board, Los Angeles, CA, USA
†Deceased

*Correspondence to*: Glenn M. Wolfe (glenn.m.wolfe@nasa.gov)

## Abstract

Large wildfires influence regional atmospheric composition, but chemical complexity challenges model predictions of downwind impacts. Here, we elucidate key connections within gas-phase photochemistry and assess novel chemical processes via a case study of the 2013 California Rim Fire plume. Airborne in situ observations, acquired during the NASA Studies of Emissions, Atmospheric Composition, Clouds and Climate Coupling by Regional Surveys (SEAC⁴RS) mission, illustrate the

evolution of volatile organic compounds (VOC), oxidants, and reactive nitrogen over 12 hours of atmospheric aging. Measurements show rapid formation of ozone and peroxyacyl nitrates (PNs), sustained peroxide production, and prolonged enhancements in oxygenated VOC and nitrogen oxides ($NO_x$).

Observations and Lagrangian trajectories constrain a 0-D puff model that approximates plume photochemical history and provides a framework for evaluating process interactions. Simulations examine the effects of 1) previously-unmeasured reactive VOC identified in recent laboratory studies and 2) emissions and secondary production of nitrous acid (HONO). Inclusion of estimated unmeasured VOC leads to a 250% increase in OH reactivity and a 70% increase in radical production via oxygenated VOC photolysis. HONO amplifies radical cycling and serves as a downwind $NO_x$ source, although impacts depend on how HONO is introduced. Addition of initial HONO (representing primary emissions) or particulate nitrate photolysis amplifies ozone production, while heterogeneous conversion of $NO_2$ suppresses ozone formation. Analysis of radical initiation rates suggests that oxygenated VOC photolysis is a major radical source, exceeding HONO photolysis when averaged over the first 2 hours of aging. Ozone production chemistry transitions from VOC-sensitive to $NO_x$-sensitive within the first hour of plume aging, with both peroxide and organic nitrate formation contributing significantly to radical termination. To simulate smoke plume chemistry accurately, models should simultaneously account for the full reactive VOC pool and all relevant oxidant sources.

## 1 Introduction

Biomass burning accounts for at least 30% of global emissions of non-methane volatile organic compounds (VOC) (Akagi et al., 2011; Andreae, 2019; Yokelson et al., 2008). Pyrogenic VOC fuel production of ozone and secondary organic aerosol, with significant consequences for air quality and climate (Jaffe and Wigder, 2012; McClure and Jaffe, 2018; Buysse et al., 2019; Val Martin et al., 2015; Hodshire et al., 2019). Pyrogenic emissions consist of thousands of unique compounds. Existing emission inventories include over 100 individual VOC (Andreae, 2019; Akagi et al., 2011). These compilations are incomplete, and it is estimated that previously "unidentified" VOC account for ~50% of total pyrogenic VOC mass (Akagi et al., 2011; Yokelson et al., 2013; Gilman et al., 2015). Recent advances in instrumentation have broadened the suite of detectable VOC to over 500 species (Sekimoto et al., 2018; Hatch et al., 2019, 2017; Koss et al., 2018), although significant uncertainty remains regarding the speciation, reactivity, and fate of this extended VOC pool.

The photochemistry of biomass burning plumes is perhaps less well understood than emissions, especially within the context of total reactive VOC. Observations of ambient smoke typically encompass a few hours of physical age and include a limited set of compounds (Akagi et al., 2012, 2013; Alvarado et al., 2010; Liu et al., 2016; Müller et al., 2016). Detailed simulations may provide some insight into the chemistry of unidentified VOC, though models are often under-constrained and both model and measurement uncertainties can be large (Lonsdale et al., 2019; Alvarado et al., 2015; Müller et al., 2016; Mason et al., 2001; Liu et al., 2016). With extended VOC inventories from recent laboratory work (Hatch et al., 2017; Koss et al., 2018), several studies have begun characterizing the impacts of previously unidentified VOC. Analyzing a series of

laboratory burns, Coggon et al. (2019) estimate that species included in the Master Chemical Mechanism (MCMv3.3.1) account for ~60% of the primary hydroxyl radical (OH) reactivity measured via Proton Transfer Time-of-flight Mass Spectrometry. This study also demonstrates that furans, a previously unconsidered class of reactive VOC, may increase ozone ($O_3$) production within an agricultural fire plume by ~10% over the first hour of aging. Decker et al. (2019) estimate that the MCM accounts for ~30% of the observed nitrate radical ($NO_3$) reactivity in the same laboratory experiments.

Despite improved knowledge of pyrogenic VOC speciation, it remains unclear how to best synthesize this information within existing regional/global model frameworks. It is not feasible to add a multitude of new species to chemical mechanisms; indeed, many global models do not explicitly account for all "known" VOC (Duncan et al., 2007; Wiedinmyer et al., 2011). Nor is it evident that such modifications would improve model results, as plume chemistry may be a "sub-grid scale" process more suited to parameterization than explicit simulation. On the other hand, reactive VOC chemistry can persist for days downwind of a fire (Mauzerall et al., 1998; Alvarado et al., 2020; Forrister et al., 2015), and an improved representation of such processes may alter regional budgets of ozone, CO, oxidized VOC, reactive nitrogen, and organic aerosol.

Oxidant sources also remain poorly understood in this environment. Smoke plumes are photochemically complex due to spatial and temporal variability in radical precursors, radiation, and other factors (Wang et al., 2021). Recent work has highlighted the importance of emitted nitrous acid (HONO) as a radical source in nascent smoke plumes (Peng et al., 2020; Theys et al., 2020; Robinson et al., 2021). Secondary HONO production via heterogeneous processes may sustain downwind radical production in some, but not all, smoke plumes (Alvarado and Prinn, 2009; Alvarado et al., 2015). Multiple mechanisms have been proposed to explain observed HONO in other environments (Zhang et al., 2019), but their controlling factors and potential impacts are not well-characterized.

Here, we utilize a case study of the 2013 California Rim Fire to examine the impacts of newly identified reactive VOC and HONO on gas-phase chemistry. Airborne in situ observations from the NASA Studies of Emissions, Atmospheric Composition, Clouds and Climate Coupling by Regional Surveys (SEAC[4]RS) mission constrain the evolution of emitted gases, oxidants, and oxidation products over ~12 hours of atmospheric aging. We combine these observations with air mass trajectories to drive a 0-D puff model that approximates plume aging. With this framework, we illustrate how an extended VOC pool and various HONO sources alter the chemistry of oxidants, oxidized VOC, reactive nitrogen, and ozone. We evaluate model results through comparison with observations and use the model to quantify age- and mechanism-dependent changes in OH reactivity, radical production/termination, and ozone production sensitivity to VOC and $NO_x$.

## 2 Methods

### 2.1 Rim Fire and SEAC[4]RS Observations

The Rim Fire was an extreme wildfire in the central Sierra Nevada Mountains of California (37.85°N, 120.08°W). Ignited by an illegal campfire on 17 August 2013, the fire was not fully contained until 24 October 2013 and consumed a total area exceeding 1000 $km^2$. At the time of SEAC[4]RS flights, the fire front spanned a width of ~40 km in both the N-S and E-W

directions. Fuels consisted mostly of mixed conifer forest (Lydersen et al., 2014). Previous Rim Fire studies have investigated

fire meteorology (Peterson et al., 2015), emissions (Liu et al., 2017; Yates et al., 2016; Saide et al., 2015), aerosol properties (Forrister et al., 2015; Adler et al., 2019; Perring et al., 2017), radiative effects (Yu et al., 2016), and regional chemistry model performance (Baker et al., 2018). We focus here on downwind gas-phase photochemistry.

SEAC$^4$RS was a NASA-funded effort to characterize the processes controlling atmospheric properties in the summertime U.S. (Toon et al., 2016). We utilize in situ observations acquired from the NASA DC-8 aircraft, which sampled Rim Fire

outflow on 26 and 27 August 2013. Analysis focuses on the "long-axis" portion of the 26 August flight, which extends from directly over the fire to 470 km downwind (Fig. 1). The actual sample time window for this leg is UTC 23:06 to 23:57, and the physical smoke age ranges from 0 to ~12 hours (Sect. 2.2). The aircraft maintained a pressure altitude of 4.3 km until 23:32 (smoke age ~5 h), when it descended to 3.6 km (Supplementary Information (SI) Fig. S1). Terrain was mountainous, and corresponding altitude above ground level ranged from 1.3 to 3.2 km. Sampling was predominantly in the upper edge of the

smoke plume, in the lower free troposphere according to boundary layer depth derived from trajectory meteorological fields (Fig. S1c). The modified combustion efficiency (MCE) near the source ranged from 0.91 to 0.94, indicating a mix of flaming and smouldering combustion. The DC-8 also sampled Rim Fire smoke as old as 2 – 3 days but influence from other fires, surface-atmosphere exchange, and changing background concentrations complicates analysis of that data. SEAC$^4$RS also investigated several other wild and agricultural fire plumes (Liu et al., 2016; Toon et al., 2016).

Table S1 lists the instruments and measurement accuracy for observations used in this study; the payload is further described in Toon et al. (2016). Here we provide a brief summary of key measurements. Most speciated VOC observations (alkanes, alkenes, aromatics, terpenes, and alkyl nitrates) derive from the Whole Air Sampler (WAS), with a sample collection time of 40 s and a sampling interval of 2 – 10 min. We also use VOC and oxygenated VOC (oVOC) observations from the Proton Transfer quadrupole Mass Spectrometer (PTR-MS), including acetaldehyde, the sum of methyl vinyl ketone and

methacrolein (MVK + MACR), and the sum of isoprene and furan. Furan is calculated as the difference between the PTR-MS sum and WAS isoprene. Formaldehyde (HCHO) is measured via both laser-induced fluorescence and infrared absorption spectroscopy. Peroxides, nitric acid, and hydroxynitrates are measured via $CF_3O^-$ chemical ionization mass spectrometry (CIMS). Other oxidized nitrogen measurements include NO and $NO_2$ via chemiluminescence; $NO_2$, total peroxy nitrates, and total alkyl nitrates via thermal dissociation – laser-induced fluorescence; and speciated peroxyacyl nitrates via thermal

dissociation iodide CIMS. Ozone is measured via chemiluminscence. Carbon monoxide (CO) is measured via differential absorption. Photolysis frequencies are calculated from observed up- and down-welling actinic flux combined with literature-recommended cross sections and quantum yields. Other observations used primarily for model inputs include pressure, temperature, water vapour (open-path absorption), particulate nitrate (aerosol mass spectrometer), aerosol surface area (laser aerosol spectrometer), and total solar irradiance. Aside from WAS data, observations are nominally reported at 1 Hz but may

contain gaps due to normal instrument operation. For this analysis, all 1 Hz data are averaged to WAS collection windows. The dataset does not include observation of total oxidized nitrogen ($NO_y$), nitrous acid (HONO), or ammonia ($NH_3$).

Normalized excess mixing ratios (NEMRs) are calculated using CO as the dilution tracer.

$$NEMR(X) = \Delta X / \Delta CO = (X - X_b)/(CO - CO_b) \qquad\qquad (1)$$

Here, X is the observed mixing ratio and $X_b$ is the background mixing ratio. Fast observations fluctuate rapidly in fire plumes

and can sometimes contain gaps over a portion of a WAS sampling interval. We adopt a custom averaging procedure to ensure rigorous NEMR calculation for such data. First, each 1 Hz variable $X$ is time-aligned to CO by applying a time lag based on the maximum cross-correlation. Second, CO is filtered to exclude points where $X$ is missing. Finally, $X$ and the filtered CO are averaged to the WAS time base. Executing this procedure over all data effectively creates a unique, gap-filtered, WAS-averaged CO for each variable. Figure S2 shows the dilution factors for each sample, ranging from 1 to 24.

Background mixing ratios are averaged over a single WAS sample collected east of the plume (orange star in Fig. 1). We also explored using observations upwind, downwind, or west of the plume for background estimation, but these samples contain stronger fire influence than the eastern sample based on the conserved fire tracers HCN and $CH_3CN$. These alternative background samples also contain higher $O_3$ (60 – 80 ppbv vs 50 ppbv), leading to smaller or negative $O_3$ NEMRs and significantly poorer agreement with the model. The influence of background selection on NEMRs depends on the relative

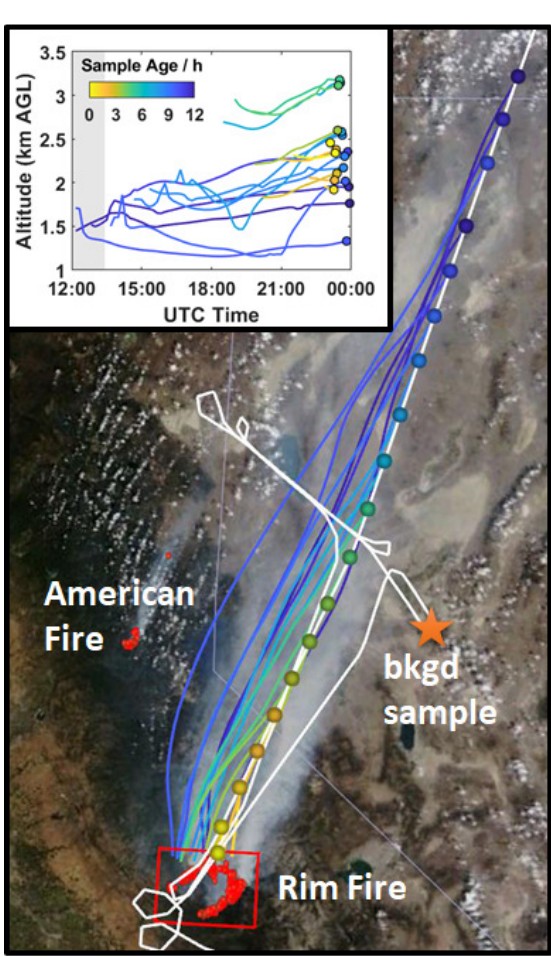

**Figure 1. Map of the study region. The background shows visible imagery and fire counts (red dots) from Terra/MODIS (https://worldview.earthdata.nasa.gov/). The white line is the DC-8 flight track. The aircraft entered from the south, flew over the fire, proceeded NE to do a missed approach at Fallon airport (near orange star), executed a "wall" pattern of perpendicular transects at several altitudes, and returned to the fire before proceeding NE along the long axis leg. Colored circles denote WAS sample locations and colored lines are the corresponding median trajectories. Yellow-to-blue shading indicates Lagrangian plume age at the time of sampling (see color bar in inset). The orange star is the location of the background sample. The red box is the "Rim Fire" box used for trajectory filtering. (Inset) Time series of median trajectory altitudes with respect to ground level. Colored circles denote the observation time. Gray shaded area indicates night time.**

magnitude of background and in-plume mixing ratios, which varies with chemical species and age. Employing a constant background introduces additional uncertainty in observed NEMRs. Use of the same backgrounds in model simulations, however, reduces the impact of such uncertainties with regard to model-measurement comparisons.

## 2.2 Trajectory Analysis

Air mass trajectories provide an estimate of smoke age and inputs for the puff model described below. Trajectories are
computed with the Hybrid Single-Particle Lagrangian Integrated Trajectory (HYSPLIT) Model (Stein et al., 2015), interfaced with custom MATLAB software for input and output handling (https://github.com/AirChem/HYSPLITcontrol, last accessed 1 Nov 2021). Trajectories are initialized at the midpoint of each WAS sample and run backward in time for 48 hours. Outputs include both position and meteorology (temperature, pressure, relative humidity, solar zenith angle (SZA), and boundary layer depth). To build statistics, we utilize archive meteorological fields from the North American Mesoscale Forecast System
(NAMs, 12 km, 1-hour) and North American Regional Reanalysis (NARR, 32 km, 3-hour) datasets (https://ready.arl.noaa.gov/archives.php, last accessed 24 March 2020). Trajectories are executed as meteorological ensembles with HYSPLIT default settings (27 members, 1 grid-point horizontal shift, 0.01 sigma unit vertical shift), giving 54 trajectories per initialization. Next, we define a geographic box for the Rim Fire based on satellite imagery and fire counts (Fig. 1). Trajectories are filtered to exclude those not passing through this box. Filtered trajectories are spatially averaged using a
geometric median (Zhong, 2021), resulting in one "median trajectory" for each observation (Fig. 1, inset). Meteorological quantities are averaged via arithmetic mean.

We define Lagrangian smoke age from the time when an averaged trajectory last intersected the Rim Fire box out to the downwind time of sampling. The standard deviation of the smoke age from individual trajectories is typically ~9% of the mean for each ensemble, and we take this as a 1σ precision estimate. Smoke at our nominal starting point (the North end of the box)
is actually a mixture of ages. Based on observed wind speed and modelled trajectories near the fire, we estimate a transit time of ~1 hour for emissions from the Southern-most fire front to reach the North end of the box. We take half of this value as an approximation for variability in actual smoke age. The total estimated uncertainty in average smoke age is then 9% + 0.5 h.

## 2.3 Puff Model

Simulation of any fire plume is challenging due to high concentrations, strong dilution, aerosol radiative perturbations, and
other factors. SEAC[4]RS did not sample the core of the Rim Fire plume, and smoke ages of 0 – 12 hours were probed over a single hour of flight. Thus, if we wish to compare with observations, it is not adequate to represent the Rim Fire as a single Lagrangian plume. Detailed fire plume models have improved in recent years (Lonsdale et al., 2019), but observational constraints are limited compared to the complexity of such a model. The goal of our simulation is to obtain a meaningful comparison against observations without over-elaboration (Box, 1976).
In our model framework, the plume is approximated as a series of 0-dimensional "puffs." The puff model is developed within the Framework for 0-D Atmospheric Modelling (F0AMv4.2.1, available at https://github.com/AirChem/F0AM/)

(Wolfe et al., 2016). One puff is simulated for each WAS sample/trajectory pair shown in Fig. 1. Each puff evolves in real time along the "average trajectory" with meteorological constraints updated every 10 minutes. For each puff, along-trajectory pressure, temperature, relative humidity, and solar zenith angle are constrained with averaged trajectory output. Trajectory meteorology is rescaled by multiplicative factors (typically within a few percent of unity) based on the ratio of observed-to-trajectory values at trajectory endpoints. Photolysis frequencies are semi-constrained to observations using a two-part scaling that helps account for smoke radiative perturbations (Appendix A). Variations in scaling factors suggest stronger attenuation in the UV than the visible (compare $J(O^1D)$ and $J(NO_2)$ in Fig. A1b), consistent with other reports (Baylon et al., 2018). Chemical concentrations from the endpoint of each puff are extracted for comparison with observations.

Initial concentrations are the same for all puffs. These are estimated by combining Rim Fire emission ratios (ERs) from Liu et al. (2017) with the excess DACOM CO mixing ratio from the first WAS sample of the long-axis leg. Normalized excess mixing ratios for the near-source WAS sample are generally within ±50% of the ERs reported by Liu et al. (2017) for the Rim Fire (Fig. S3). The Liu et al. (2017) ERs incorporate multiple near-source intercepts and are more representative of average fire conditions. Actual ERs likely vary among the plume samples. Older samples represent fire emissions from earlier in the day, and we might expect these samples to show an increased signature of smouldering relative to flaming due to typical wildfire diurnal progression (Wiggins et al., 2020). Observations, however, do not conclusively indicate a time-dependent trend in ERs. MCE generally declines with age (Fig. S4a), consistent a shift from smouldering to flaming over time, but frequent deviation from the expected wildfire value range of 0.8 – 1.0 (Akagi et al., 2011) suggests non-emission influence (e.g., background $CO_2$ variability) that degrades this metric as a combustion phase tracer at later ages. NEMRs for the conserved fire tracers HCN and $CH_3CN$ exhibit some variability (standard deviation/mean = 10 – 15%) but no trend (Fig. S4b), suggesting no systematic change in fire-average ERs (Roberts et al., 2020). Conversely, the trend in total observed $NO_y$ may indicate time-varying emissions (Sects. 3.1.4 and 4.2). Given these ambiguities, constant initial concentrations is a reasonable assumption.

Dilution is treated in analogy with Gaussian plume dispersion (Alvarado et al., 2015):

$$\left(\frac{\partial X}{\partial t}\right)_{dil} = \frac{-1}{\tau_g + 2t}(X(t) - X_b) \tag{2}$$

Background concentrations ($X_b$) are the same as those used for dilution normalization in Eq. 1. The Gaussian timescale, $\tau_g$, is constant for each puff and calculated using the analytical solution of the integral of Eq. (2) with CO concentrations at start and end points. The Gaussian timescale varies from 28 to 760 s for individual puffs (Fig. S5). Such variability is not surprising given the horizontal extent of the fire and differences among trajectories for each puff.

Model chemistry utilizes the Master Chemical Mechanism (MCMv3.3.1) (Jenkin et al., 2015, 1997; Saunders et al., 2003) with modifications. Additional reactions include photolysis of pernitric acid (Atkinson et al., 2004), reaction of methyl peroxy radical with OH (Assaf et al., 2016; Caravan et al., 2018), reaction of hydroxymethyl hydrogen peroxide (HMHP) with OH (Allen et al., 2018), and oxidation of propadiene ($C_3H_4$). For the latter, we use the OH reaction rate coefficient of Atkinson and Arey (2003) rather than the 1.8-times slower rate coefficient of Daranlot et al. (2012), based on the similar observed decay

rates of propadiene and ethene. Subsequent propadiene chemistry follows the mechanisms of Daranlot et al. (2012) and Xu et al. (2019). We also update rate coefficients for reaction of peroxyacetic acid with OH (Berasategui et al., 2020) and peroxyacyl radicals with $HO_2$ (Jenkin et al., 2019), which are slower/faster than MCM default values by factors of 123/1.33, respectively. Oxidation of some additional biomass burning VOC (furans, syringol, and guaiacol) is incorporated using an MCM extension developed following recent laboratory and field studies (Coggon et al., 2019; Decker et al., 2019, 2021a; Robinson et al., 2021).

There are multiple potential error sources in the puff model, and many are not easily quantified. We assume constant initial concentrations, but the smoke is a heterogeneous mixture of multiple ages and burning phases. Parameterizations for photolysis and dilution make assumptions about the history of each puff based on the observed evolution along the flight path. Heterogeneous chemistry is not explicitly included. Despite all these issues, results will illustrate that the puff model is a reasonable approximation and a useful testbed for probing plume chemistry. Measurement accuracy (Table S1) is typically the dominant uncertainty in observations at high signal/noise ratios (rather than precision), thus we use this to define the uncertainty for model-measurement comparisons.

## 2.4 Model Scenarios

Simulations systematically characterize the effects of varying emissions and chemistry (Table 1). In the base simulation (M0), initial species are limited to observations. This includes three species that are not in the MCM: furan, HMHP, and propadiene. Additional simulations incorporate unmeasured reactive VOC and HONO sources as detailed below.

**Table 1. Summary of model simulations.**

| Simulation | Description |
|---|---|
| M0 | Base simulation using only measured VOC |
| M1 | M0 + Unmeasured VOC |
| M2a,b,c | M1 + primary HONO (5, 15, 25 ppbv) |
| M3a,b,c | M1 + secondary HONO via $pNO_3^-$ photolysis ($J(pNO_3^-)$ scaling factors of 0.5, 1, 2) |
| M4a,b | M1 + secondary HONO via $NO_2$ + aerosol ($\gamma(NO_2)$ scaling factors of 1, 1000) |

## 2.4.1 Addition of Unmeasured VOC

Simulation M1 incorporates unmeasured VOC using the fire laboratory emissions data of Koss et al. (2018), which includes over 500 compounds and 20 Western US fuel types. We restrict this dataset to 152 compounds with specific molecular assignments and estimated OH reaction rate coefficients (Table S5 of Koss et al. (2018)). Incorporating this data requires 1) estimating initial VOC concentrations and 2) allocating these VOC to model species. For the first step, we create a composite ER profile for the Rim Fire by comparing the ERs of 11 species reported by both Liu et al. (2017) and Koss et al. (2018):

acetylene, ethene, propene, methanol, formaldehyde, acetaldehyde, furan, benzene, toluene, HCN, and $CH_3CN$. Specifically, we optimize fractional fuel contributions by minimizing the sum square relative error of log-transformed ERs between Liu et al. (2017) and the composite profile. Estimated fuel composition is sensitive to which VOC are included in the optimization, as ER profiles are highly correlated among different fuels. For the same reason, model results are not especially sensitive to the choice of fuel composition. Combining composite ERs with observed CO in the first plume sample gives initial mixing

ratios. We assume zero background for all unmeasured VOC. ERs for acrolein and biacetyl are reduced by factors of 2.3 and 10, respectively, due to known calibration issues (SI Text S1).

Unmeasured VOC are assigned to model species via one of two methods. 51 species appear in the MCM or the extended NOAA biomass burning mechanism and are thus accounted for explicitly. The remaining 101 compounds lack direct MCM analogues. These VOC are allocated to MCM proxies based on OH reaction rate coefficients ($k_{OH}$) and molecular formulae.

For each VOC, we first identify MCM species that are within some threshold (nominally 20%) of the VOC's estimated $k_{OH}$. We then attempt to filter the MCM species list to include only those containing a similar number of carbon and oxygen atoms (within $\pm 1$ the number of each atom). Specific functional groups are not considered. If the molecular formulae criterion is too restrictive (i.e., no species are identified), we only use the $k_{OH}$ criterion. Table S2 lists the MCM assignments and ERs for all unmeasured VOC. Figure S6 compares bulk chemical metrics between the non-MCM species of Koss et al. (2018) and

assigned MCM proxies. Overall, proxies reproduce the distribution of OH reactivity (total 47 $s^{-1}$) and carbon content (390 ppbv C) and are biased high with respect to oxygen content and molecular weight. This is not surprising, as most VOC contained in the MCM are multi-generation oxidation products..

### 2.4.2 Addition of HONO

HONO was not measured during SEAC[4]RS. Several sets of sensitivity simulations explore the impacts of primary (emitted)

and secondary HONO. All of these simulations include the same initial unmeasured VOC as simulation M1. Simulations M2a, b, and c incorporate primary HONO at initial mixing ratios of 5, 15, and 25 ppbv, respectively. The upper end of this range is based on recent work reporting an average $\Delta HONO/\Delta NO_x$ emission ratio of $0.7 \pm 0.3$ ppbv ppbv$^{-1}$ and an average $\Delta HONO/\Delta CO$ emission ratio of $5.3 \pm 5.2$ pptv ppbv$^{-1}$ for Western U.S. wildfires (Peng et al., 2020).

Simulations M3a, b, and c incorporate photolysis of particulate nitrate ($pNO_3^-$):

$$pNO_3^- + h\nu \rightarrow 0.67HONO + 0.33NO_2 \tag{R1}$$

The photolysis frequency for reaction R1 is calculated as 286*J($HNO_3$) following Ye et al. (2017, 2018). The mechanism for this reaction is not well understood (Baergen and Donaldson, 2013), and the efficacy of $pNO_3^-$ photolysis likely depends on aerosol composition (Ma et al., 2021). Other studies have estimated photolysis frequencies an order of magnitude or more lower in non-biomass burning environments (Romer et al., 2018; Shi et al., 2021). Particulate nitrate concentrations are

constrained by aerosol mass spectrometer (AMS) observations (Fig. S7a). Comparison with other aerosol composition observations suggests minor $pNO_3^-$ contained in coarse mode aerosol, which is excluded by the AMS (Fig. S8). It is not possible to partition AMS $pNO_3^-$ between organic and inorganic forms for SEAC[4]RS (Ulbrich et al., 2009; Day et al., 2021);

however, the nature of particulate nitrate participating in reaction (R1) is also unclear. For all puffs, $pNO_3^-$ is initialized with the observed concentration in the near-source sample. The model treats $pNO_3^-$ as non-reactive (no chemical production or loss), but it dilutes with a puff-dependent background concentration, chosen such that concentrations match observations at the start and end of each puff. This is a workaround to ensure the model carries reasonable $pNO_3^-$ concentrations throughout the simulation in the absence of explicit model aerosol chemistry. Simulations M3a, b, and c scale the nominal rate of reaction (R1) by factors of 0.5, 1, and 2, respectively.

Simulations M4a and b incorporate reaction of $NO_2$ on aerosol surfaces:

$$NO_2 \xrightarrow{k_a} HONO \tag{R2}$$

$$k_a = 0.25 v_{NO2} S_a \gamma \tag{3}$$

Here, $k_a$ is the first-order rate coefficient, $v_{NO2}$ is the mean molecular speed of $NO_2$, $S_a$ is particle surface area density, and $\gamma$ is the reactive uptake coefficient. We assume that gas diffusion is not a limiting factor for small values of $\gamma$ ($10^{-3}$ to $10^{-6}$). For $S_a$, we use Laser Aerosol Spectrometer observations (Yu et al., 2016) scaled up by a factor of 1.7 to account for calibration bias (Fig. S8) and linearly interpolated over plume age. Values of $S_a$ range from 800 to 6600 $\mu m^2\ cm^{-3}$ (Fig. S7b). Parameterization of $\gamma$ follows a laboratory-derived relationship with solar radiation (Stemmler et al., 2006; Zhang et al., 2019):

$$\gamma = \begin{cases} \alpha J_{NO2}, & Rad \leq 400\ W\ m^{-2} \\ \alpha J_{NO2}(Rad/400)^2, & Rad > 400\ W\ m^{-2} \\ 10^{-6}, & minimum \end{cases} \tag{4}$$

Here, $J_{NO2}$ is the $NO_2$ photolysis frequency derived from the model, $\alpha = 2.5 \times 10^{-4}$ is a scaling factor, and *Rad* is total solar irradiance. The latter is estimated using trajectory-dependent SZA and the linear relationship between SZA and observed solar irradiance (Fig. S9). Some minor systematic bias in *Rad* may result from extrapolation of this relationship. The minimum value of $\gamma$ follows Aumont et al. (2003). Figures S7c-d show calculated $\gamma$ and $k_a$ at the end point of each puff. The uptake coefficient ranges from $4 - 11 \times 10^{-6}$, while $k_a$ ranges from $0.3 - 4.9 \times 10^{-6}\ s^{-1}$. Note that $k_a$ varies along the trajectory for each puff due to changes in radiation and $S_a$. Simulation M4a uses this default parameterization, while in simulation M4b $k_a$ is multiplied by a factor of 1000. This range spans observed $NO_2$ uptake on humic acid ($\gamma = 2 - 8 \times 10^{-5}$ (Stemmler et al., 2006)) and soot ($\gamma = 3.7 - 11 \times 10^{-3}$ (Ammann et al., 1998)) surfaces. We do not consider ground surface HONO sources (Chai et al., 2021), as observations are limited to the free troposphere and uppermost mixed layer.

## 3. Results

### 3.1 Observations and Base Simulation

We first examine the evolution of observed trace gases in the Rim Fire plume. Observations illustrate several general features, including 1) rapid oxidation of primary emissions and production of secondary species in the first 2 h of aging, 2) mixing with biogenic emissions around an age of 2 – 3 h, and 3) a transition in some species associated with a decrease in sampling altitude

around an age of 6 h (Fig. S1). The following sections survey age-dependent trends in primary VOC, peroxides, oxygenated VOC, reactive nitrogen, and ozone. Comparison to base simulation output (M0) benchmarks the representation of this chemistry with MCMv3.3.1 and available constraints.

### 3.1.1 VOC

The downwind evolution of primary VOC illustrates the differing effects of emissions, dilution, and oxidation. We group VOC into four categories: long-lived alkanes, aromatics/intermediate-lived alkenes, short-lived alkenes, and biogenic terpenes. Groups reflect photochemical lifetimes and similarities in model NMB. Figures 2a–d show an example from each category, Fig. S10 shows the time series of all observed VOC, and Fig. S11 shows the normalized mean bias (NMB) for each VOC calculated over the full simulation.

Long-lived alkanes such as ethane, propane, and butanes do not decay monotonically with plume age (Figs. 2a and S10a–d). NEMRs for these gases peak downwind of the Rim Fire; for example, the highest propane NEMR occurs at an age of 4 h and is 50% higher than the initial value. All model simulations exhibit negative NMB of –26% or less for these gases (Fig. S11). With lifetimes of days or more against OH oxidation, these gases are relatively sensitive to variability in background levels and regional emissions (e.g., urban and oil/natural gas activities). For example, at an age of 4 h the ethane enhancement is 4 times its background value, whereas the ethene enhancement is a factor of 70. Variability in fire emissions is a less likely explanation for model-measurement mismatch. Reconciliation with the observed increase in NEMRs at an age of 2 h would require VOC-to-CO emission ratio increases of 50 – 100%, and we would expect similar changes in other VOC emission ratios (Liu et al., 2017; Permar et al., 2021) that are not observed. These VOC are minor contributors to plume photochemistry; however, this comparison underscores the challenge of accounting for background variability in Lagrangian or pseudo-Lagrangian simulations.

The second group, including most aromatics (benzene, toluene, ethyl benzene and xylenes) and several alkenes (ethene and propadiene), exhibits stronger NEMR decays with age (Figs. 2b and S10e–k). Photochemical lifetimes for these gases range from 11 – 48 hours (except for benzene, with a lifetime of 4 – 10 days), thus we expect both emissions and chemistry to influence NEMR variability. Similar to alkanes, longer-lived aromatic NEMRS peak at 2 – 6 hours of age. Xylenes, ethene and propadiene exhibit a more monotonic decay. The base model NMB is below 8% for the latter three compounds.

Oxidation controls the trend of short-lived alkenes, including propene, butenes, butadiene, pent-1-ene, and furan (Figs. 2c and S10l–s). These gases have near-zero background mixing ratios and react primarily with OH. Loss via $O_3$ reaction is < 15% of the total loss rate, and nitrate radical reaction is negligible within sampled air masses. NEMRs decay rapidly, and the most reactive gases fall below measurement detection limits at later ages. The base model simulation exaggerates this decay, resulting in negative NMB as large as -233% for furan.

Terpenes (isoprene and α/β-pinene) are also highly reactive, and NEMRs generally decay rapidly (Figs. 2d and S10t,u). Around an age of 2 h, however, NEMRs increase beyond those in the near-source sample. A similar pattern appears

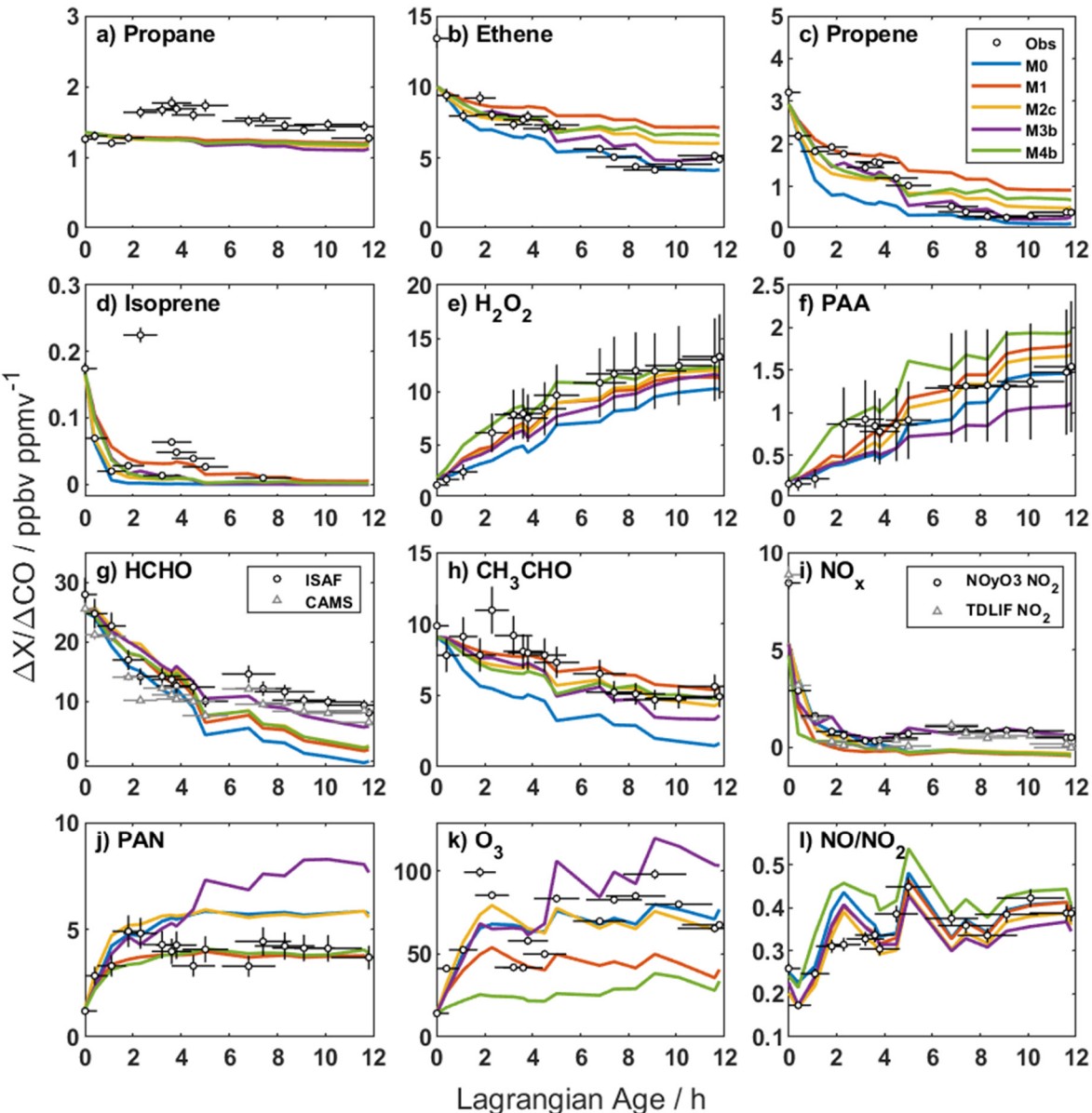

**Figure 2. Age evolution of NEMRs for reactive gases (a-k) and the NO/NO₂ ratio (l). Black circles are observations with their corresponding uncertainty due to measurement accuracy and age. Colored lines are model output from the base simulation (M0, blue), addition of unmeasured VOC (M1, red), and addition of unmeasured VOC plus initial HONO (M2c, yellow), pNO₃⁻ photolysis (M3b, purple), or NO₂ heterogeneous uptake (M4b, green). In (g) and (i), circles and triangles represent two independent measurements of HCHO and NO₂, respectively (Table S1). NO and NO₂ observations in (l) are from the NOyO3 instrument.**

in the isoprene oxidation products methyl vinyl ketone (MVK) and methacrolein (MACR) (Fig. S12d). Furthermore, this particular sample is uncharacteristically dilute relative to the Lagrangian age (Fig. S2). This evidence suggests that local biogenic emissions mixed with the Rim Fire smoke samples at Lagrangian ages of 2 – 4 hours.

### 3.1.2 $HO_x$ and Peroxides

The SEAC⁴RS DC-8 payload did not include observations of $HO_x$ (= OH + $HO_2$). The loss and production of other compounds, however, indirectly constrains $HO_x$ abundance. The decay rate of short-lived alkenes provides a benchmark for OH, while peroxide production is an indicator for $HO_2$ and, to some extent, organic peroxy radicals ($RO_2$).

As discussed above, short-lived alkene NEMRs decay faster than observed NEMRs in the base simulation, especially in the first few hours. The discrepancy between modelled and observed short-lived alkenes suggests over-prediction of OH in 345    the young plume. Base simulation OH starts at $4.9 \times 10^6$ cm$^{-3}$ and declines to $\sim 1.2 \times 10^6$ cm$^{-3}$ after $\sim 4$ hours (Fig. S13a).

Self-reaction of $HO_2$ produces hydrogen peroxide ($H_2O_2$), and reaction of $HO_2$ with peroxyacetyl radical (PA) produces peroxyacetic acid (PAA). NEMRs for both peroxides start low and increase with plume age (Fig. 2e,f). Initial mixing ratios are near background levels, at odds with a previous study suggesting significant primary emissions (Lee et al., 1997). The base simulation exhibits an upward trend but with a slower growth rate, especially over the first 4 hours. Model $H_2O_2$ 350    NEMRs skirt the lower edge of measurement accuracy, while PAA agreement generally improves with age. This comparison suggests under-prediction of $HO_2$ and possibly $RO_2$ at early ages. Base model $HO_2$ peaks at $\sim 85$ pptv and declines to values as low at 23 pptv (Fig. S13b).

### 3.1.3 Oxygenated VOC

Formaldehyde (HCHO), an oxidation product of numerous VOC, serves as a top-down constraint on in-plume VOC 355    processing. The observed HCHO NEMR decreases by a factor of three in the first 5 hours before levelling out at later ages (Fig. 2g). This behavior is consistent with loss of primary emissions alongside secondary production (Liao et al., 2021). The HCHO lifetime due to photolysis and OH oxidation is $\sim 3.5$ h at the time of sampling, but accumulation of multi-generation oxidation products may sustain downwind HCHO production (Alvarado et al., 2020). The rise in the HCHO NEMR between 5 and 7 hours coincides with descent to a lower sampling altitude (Fig. S1). The base simulation follows observations early on 360    but underestimates NEMRs later.

The acetaldehyde ($CH_3CHO$) NEMR decay exhibits a more constant slope (Fig. 2h). The small post-emission peak in the acetaldehyde NEMR at an age of 2 h coincides with the aforementioned sharper peaks in biogenic markers and enhancements in long-lived alkanes, which may indicate influence from non-fire surface emissions. Base model NEMRs initially decay more rapidly than observed, but the slope matches observations after the first few hours. This discrepancy is 365    consistent with over-prediction of OH in the young plume, as OH oxidation accounts for 80% of acetaldehyde loss. Conversely,

OH only accounts for 15 – 50% of HCHO loss (photolysis is the remainder). Thus, as will be evident in sensitivity simulations discussed later, HCHO and $CH_3CHO$ exhibit opposite responses to varying OH.

Text S2 and Figure S12 present several other oxygenated VOC (oVOC) observations, including methanol, acetone + propanal, hydroxyacetone, and MVK + MACR.

### 3.1.4 Reactive Nitrogen

Major wildfire reactive nitrogen emissions include $NO_x$, HONO, and $NH_3$ (Lindaas et al., 2020; Roberts et al., 2020). Observations of HONO and $NH_3$ are not available for $SEAC^4RS$, so here we focus on $NO_x$ and observed reservoir species including peroxyacyl nitrates, alkyl nitrates, and nitric acid.

The base simulation reasonably reproduces the initial loss of $NO_x$ but does not capture later behavior (Fig. 2i). In the first few hours, the $NO_x$ NEMR decays with an e-folding timescale of 20 – 40 minutes. After reaching a minimum at ~3.6 h, the $NO_x$ NEMR rises to a sustained enhancement at ~10% of the initial value. An additional $NO_x$ source of 200 pptv $h^{-1}$ is required to close the $NO_x$ budget in the period between 6 and 12 h. This discrepancy is explored further in Sect. 3.3.

Peroxyacetyl nitrate (PAN) typically comprises the majority of total peroxy nitrates (Wooldridge et al., 2010) and is produced via the reversible reaction of $NO_2$ with PA. The PAN NEMR rises rapidly in the first two hours before stabilizing (Fig. 2j), comparable to previous observations (Alvarado et al., 2010, 2015). The base simulation captures the early rise in PAN but overshoots the asymptote by ~20%. SI Text S3 and Figures S14-S15 compare model output to measurements of other speciated peroxy nitrates, total peroxy nitrates, and alkyl nitrates, which show varying levels of model-measurement agreement.

Reaction of OH with $NO_2$ primarily forms nitric acid ($HNO_3$). The observed $HNO_3$ NEMR is negative and trends downward with age, indicating that in-plume $HNO_3$ is below background levels (Fig. S15c). Excess ammonia/ammonium (Perring et al., 2017) likely drives rapid formation of particulate ammonium nitrate (Lindaas et al., 2021). The puff model lacks aerosol chemistry and thus predicts net growth of the $HNO_3$ NEMR. The difference between the base model and observed $HNO_3$ NEMR implies an effective $HNO_3$ lifetime of less than 1 hour within the plume.

Figure 3 shows the evolution of the sum of observed $NO_y$. We refer to this quantity as $\Sigma NO_{y,obs}$ to acknowledge missing observations of some $NO_y$ species such as HONO, $HO_2NO_2$, nitroaromatics, and possibly other organonitrates. PNs comprise 20% of $NO_{y,obs}$ within the first sample, consistent with a mixture of smoke ages. After the first hour, PNs comprise more than half of $\Sigma NO_{y,obs}$, $pNO_3^-$ comprises another 20 – 30%, and $NO_x$ and alkyl nitrates contribute 10 – 15% each. The $\Sigma NO_{y,obs}$ NEMR decreases from as high as 19.9 ppbv $ppmv^{-1}$ to as little as 6 ppbv $ppmv^{-1}$ over 12 hours of aging. Underlying this trend is a rapid $NO_x$ decay, a step-change in $pNO_3^-$ at an age of 2 h, and a gradual decline in $\Sigma PNs$ (Fig. S16). Figure S17 compares observed gas-phase $NO_y$ ($\Sigma NO_{y,gas}$, which excludes $pNO_3^-$) with the model-equivalent sum. The observed gas-phase sum decreases over age from 12.5 to 4.4 ppbv $ppmv^{-1}$ and is ~7 ppbv $ppmv^{-1}$ throughout simulation M0. The causes and consequences of this discrepancy are further discussed in Sect. 4.1.

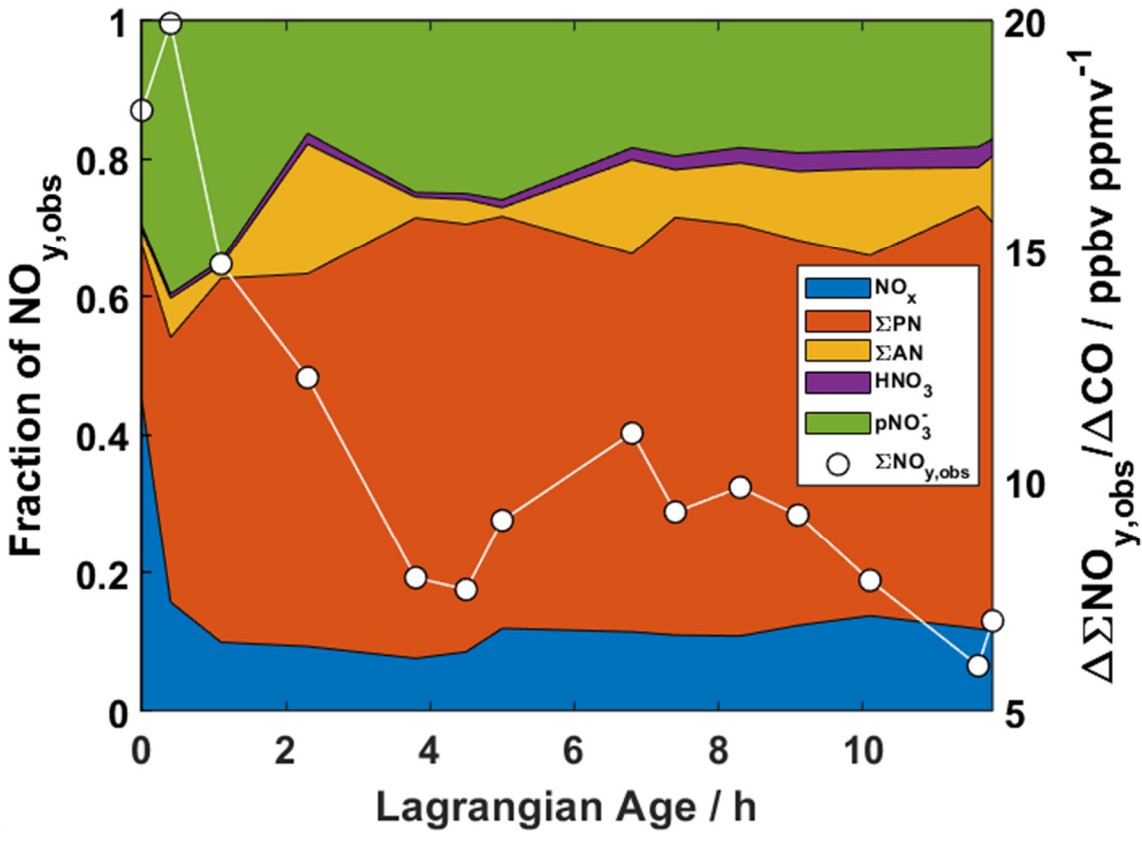

**Figure 3. Age evolution of observed NO$_y$ speciation (solid colors, left axis) and the ΣNO$_{y,obs}$ NEMR (white circles/line, right axis). ΣNO$_{y,obs}$ is the bottom-up sum of observed NO$_y$. ΣPN represents TDLIF observations.**

### 3.1.5 Ozone

The observed O$_3$ NEMR grows rapidly within the first few hours before slowing down (Fig. 2k). The peak value at 2 h coincides with the aforementioned peak in biogenic VOC. The base simulation reproduces the general trend of the observed O$_3$ NEMR time profile but misses the maximum and under-predicts after 5 h. Significant point-to-point variability in the observed O$_3$ NEMR mostly reflects variability in the CO excess mixing ratio rather than O$_3$ itself, as O$_3$ is relatively close to its background value. Absolute O$_3$ reaches a peak of 130 ppbv within the first hour before diluting rapidly to 65 – 75 ppbv, remaining above

the background estimate of 50 ppbv (Fig. S18). Mean absolute bias for base simulation O$_3$ is -3.2 ppbv averaged over all observations and +0.5 ppbv for ages greater than 2 h.

The ratio of NO to NO$_2$ relates closely to radical turnover and O$_3$ production. In the absence of strong NO$_x$ sources or sinks, photolysis of NO$_2$ and oxidation of NO establishes a photostationary state that controls this ratio:

$$\frac{[NO]}{[NO_2]} = \frac{J_{NO2}}{k_{NO+O3}[O_3] + k_{NO+HO2}[HO_2] + \sum k_{NO+RO2}[RO_2])} \qquad (5)$$

Here, $J_{NO2}$ is the $NO_2$ photolysis frequency and $k_{X+Y}$ are reaction rate coefficients. The observed $NO/NO_2$ ratio doubles over the course of 12 hours (Fig. 2l), consistent with the decline of peroxy radicals and ozone mixing ratios. $J(NO_2)$ does not exhibit a clear trend over this period (Fig. A1). The base simulation over-predicts this ratio in the first few hours and agrees within uncertainties afterward. Disagreement at young ages is consistent with insufficient conversion of NO to $NO_2$, possibly due to insufficient ozone and/or peroxy radicals.

### 3.2 Accounting for Unmeasured VOC

Differences between observations and the base simulation are consistent with missing reactive VOC in the model. The decay of reactive alkenes is faster than observed, suggesting that model OH is too high. Production of peroxides and HCHO is too slow, indicating missing sources of peroxy radicals and organic carbon. Simulation M1 approximates the effects of unmeasured VOC, incorporated following the procedures outlined in Sect. 2.4.1.

Total OH reactivity – the inverse of the OH lifetime – increases significantly upon addition of unmeasured VOC (Fig. 4a). Initial OH reactivity grows from 77 to 182 $s^{-1}$. The top five components of OH reactivity in the observations and simulation M0 are HCHO, CO, $CH_3CHO$, furan, and propene (Fig. 4c). Species included in simulation M0 comprise ~45% of the OH reactivity in simulation M1. The top 5 additional contributors in simulation M1 are aromatics, consistent with Coggon et al. (2019). Enhancements persist as the plume ages, with OH reactivities of 7 $s^{-1}$ (M1) versus 3 $s^{-1}$ (M0) at an age of 12 h. After 12 h, 32% of M1-simulated OH reactivity is comprised of over 2100 species that are, individually, not very abundant (Fig. 4c, grey area). Comparing M0 and M1 suggests that 85% of this "secondary" reactivity (1.8 $s^{-1}$) is due to oxidation of unmeasured VOC.

Normalizing for dilution reveals a more modest decline in OH reactivity due to photochemistry alone (Fig. 4b). Normalized excess OH reactivity declines by 61% and 45% for M0 and M1, respectively. The relatively slower decline in M1 reflects less OH in this simulation. Modelled declines in OH reactivity over 12 h of aging imply pseudo-first order lifetimes of 12.7 h and 21.4 h for OH reactivity in simulations M0 and M1, respectively. OH reactivity estimates from the model may be an upper limit, as some reactive carbon will partition to the aerosol phase (Palm et al., 2020).

Additional VOC reactivity markedly alters downwind chemistry. OH decreases to $5 \times 10^5$ $cm^{-3}$ for most of the simulation (Fig. S13a), reducing the decay of reactive alkenes to a shallower slope than observed (Fig. 2b–d). Maximum $HO_2$ increases by 35% (Fig. S13b), and $H_2O_2$ agrees better with observations (Fig. 2e). PAA increases due to enhanced production of $HO_2$ and PA (Fig. 2f). Peroxyacetyl radical is also a PAN precursor, but PAN actually decreases in simulation M1 due to lower $NO_2$ (Fig. 2j) and competing formation of larger peroxy nitrates (Fig. S14, S15). Rapid sequestration of $NO_x$ also suppresses ozone formation in the young plume (Fig. 2k). HCHO and $CH_3CHO$ increase (Fig. 2g–h), but the model still under-predicts HCHO at later ages.

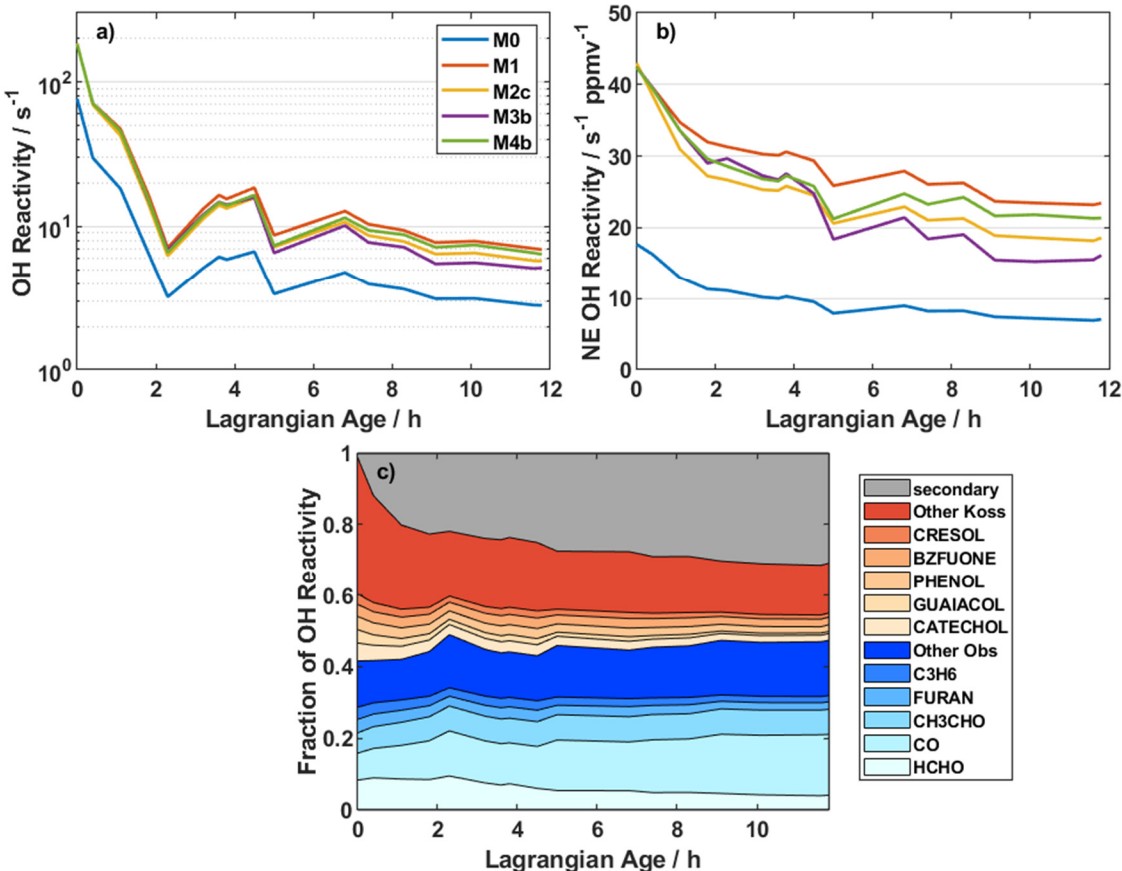

**Figure 4. (a)** Age evolution of total model OH reactivity, **(b)** normalized excess OH reactivity, and **(c)** fractional contributions of individual reactants to total OH reactivity in simulation M1. In (b), normalization is analogous to NEMR calculation (Eq. 1), and the OH reactivity background is 1.1 s⁻¹ based on summation over observed OH reactants in the background sample. In (c), blue shades denote measured compounds, red shades are species added from the Koss et al. (2018) inventory, and grey includes 2,116 other MCM species reacting with OH.

## 3.3. Accounting for Primary and Secondary HONO

The contrast of observations and simulation M1 in Fig. 2 implies insufficient OH and $NO_x$. The decay of short-lived alkenes is now slower than observed owing to the large increase in OH reactivity, while the decay of $NO_x$ is faster than observed due to rapid peroxy nitrate formation. Faster $NO_x$ sequestration also reduces $O_3$ production.

Potential explanations for model-measurement discrepancies in $NO_x$ include $NO_2$ measurement artifacts, emissions variability, unaccounted-for recycling via $NO_x$ reservoirs, and missing $NO_x$ sources. $NO_2$ measurement artifacts are unlikely to play a significant role, given the excellent correlation between two independent $NO_2$ measurements and the magnitude of model-measurement disagreement (Text S3). Doubling initial $NO_x$ has a minor effect on the simulated $NO_x$ NEMR after ~5 h (Fig. S19), and total observed $NO_y$ decreases with Lagrangian age (Fig. 3); thus, $NO_x$ emissions variability also cannot explain

this difference. Recycling via decomposition or oxidation of $NO_x$ reservoirs, such as organic nitrates, should be adequately

captured by simulation M1 as the NEMRs of total peroxy nitrates and alkyl nitrates are simulated well (Fig. S14 and S15). We lack observations of $HO_2NO_2$ or $CH_3O_2NO_2$, but lifetimes for these gases are a few minutes for model conditions and modelled mixing ratios are less than 10 pptv after 5h. Nitroaromatic photolysis may generate HONO (Sangwan and Zhu, 2016), but these compounds do not build up to sufficient levels to act as a major $NO_x$ source in the model. Therefore, a missing $NO_x$ source is the most likely explanation.

Previous work has demonstrated the important role of HONO in smoke plume chemistry (Alvarado et al., 2015; Alvarado and Prinn, 2009; Peng et al., 2020; Theys et al., 2020). Direct HONO emissions amplify radical production in the nascent plume, while secondary HONO formation may sustain oxidation as the plume ages. As discussed in Sect. 2.4.1 and summarized in Table 1, we implemented both primary emissions (simulations M2a,b,c) and secondary production via $pNO_3^-$ photolysis (M3a,b,c) or heterogeneous $NO_2$ reaction (M4a,b). Figure S20 shows the absolute HONO mixing ratio, HONO

NEMR, and $HONO/NO_2$ ratio for several representative simulations. The HONO photolysis lifetime, based on observed photolysis frequencies, is 10 – 20 minutes. HONO mixing ratios in simulations with secondary production (M3b and M4b) are 10's of pptv after a few hours. Initial HONO NEMRs and $HONO/NO_2$ ratios fall within the range observed in other fire plumes (Peng et al., 2020; Theys et al., 2020). Most figures in the main text and supplement display results from simulations with high primary HONO (M2c), moderate $pNO_3^-$ photolysis (M3b) and fast $NO_2$ heterogeneous uptake (M4b), while Figures S21 – S23

show full results for each sensitivity series.

Initial HONO stimulates chemistry in the first several hours. Simulation M2c starts with 25 ppbv of HONO, comparable to maximum levels observed in other Western U.S. wild fires (Peng et al., 2020). Photolysis of HONO increases OH and NO production, leading to faster VOC decay and product formation (Fig. 2 and S11). Intensification is limited to the first few hours, and the decay of short-lived alkenes, like propene and butenes, is too slow at later ages (Fig. 2c). $NO_x$ agreement

also improves at early times, but under-prediction persists after 3 h of aging (Fig. 2i). PAN and $O_3$ profiles are close to those in the base simulation (Fig. 2j-k). Observed $NO_y$ NEMRs are over-predicted (Fig. S17).

Photolysis of $pNO_3^-$ leads to more sustained impacts. Simulation M3b uses the literature-derived rate for reaction R1 (Ye et al., 2017). For this case, median OH increases by a factor of 3.5 relative to simulation M1 (Fig. S11), improving model agreement with short-lived VOC (Fig. 2b-c). $HO_2$ is mostly unchanged, reflecting a counterbalance of faster production via

VOC oxidation and faster loss via reaction with NO. Unique to the mechanisms tested in this study, $pNO_3^-$ photolysis reproduces the enhancements in HCHO and $NO_x$ NEMRs observed at later ages (Fig. 2g, 2i). The HONO NEMR at ages beyond 2 h also aligns with values observed in other wildfires (Fig. S20b). On the other hand, PAN and ozone are now over-predicted (Fig. 2j-k), as are $\Sigma PN$, some speciated PNs, and $\Sigma NO_{y,obs}$ (Fig. S14, S15, S17). Halving the $pNO_3^-$ photolysis rate improves agreement with PAN and $O_3$ at the expense of $NO_x$ and HCHO (Fig. S22, compare simulations M3a and M3b). Over-

prediction of PAN and its analogues in these simulations may reflect errors in the VOC distribution and/or chemical kinetics (Sect. 4.3). These simplified simulations also do not consider age-dependent variability in aerosol composition (e.g., organic versus inorganic $pNO_3^-$), which could influence the effective $pNO_3^-$ photolysis rate and product yields.

The supply of $pNO_3^-$ imposes a practical limit on the rate of HONO and $NO_2$ production via this process. Particulate nitrate production and loss is not rigorously modelled in our simulations, but we can estimate the magnitude of this limitation. A linear fit to the observed $pNO_3^-$ NEMR yields an e-folding timescale of 9.6 h, whereas the effective lifetime of $pNO_3^-$ with respect to photolysis in simulation M3b is $3.4 \pm 0.6$ h (mean and standard deviation, averaged over the endpoints of all puffs). Nitric acid and organic nitrate partitioning to particles may resupply some $pNO_3^-$ downwind; observations in other wildfire plumes show $pNO_3^-$ NEMRs increasing with age (Juncosa Calahorrano et al., 2020). Given uncertainties regarding the fate of $NO_y$ (Sect. 4.1), the observed $pNO_3^-$ lifetime is not a strong constraint on the potential chemical loss; however, $pNO_3^-$ photolysis rates are unlikely to be larger than those used in our study, and the comparison with observed gas-phase $NO_y$ (Fig. S17) suggests they may be substantially slower.

HONO production via heterogeneous reaction of $NO_2$ generally degrades agreement with observations (Fig. S23). Results from simulation M4a are nearly identical to those from M1, while in simulation M4b ($\gamma$ multiplied by 1000) ozone and $NO_x$ under-prediction worsens. Coupled with rapid HONO photolysis, this process effectively converts $NO_2$ to NO while generating OH. This acts as a non-photolytic $NO_2$ sink and increases the loss of $O_3$ to reaction with NO (reflected in the $NO/NO_2$ ratio), reducing net $O_3$ production. $NO_2$ conversion to HONO is nearly $NO_x$-neutral, whereas $pNO_3^-$ photolysis is effectively a $NO_x$ source (Fig. S17). This is also evident in the $HONO/NO_2$ ratio, which ranges from 0.2 to 3.9 and exceeds the ratio for simulation M3b by a factor of 4 or more (Fig. S20c).

Additional reactive VOC and HONO chemistry can collectively improve model-measurement agreement for most observed species, but the implementation of each process is not quantitatively independent. Model performance relative to observations inherently relies on a balance between oxidant sinks (VOC) and sources (HONO), both of which are uncertain. SI Text S4 describes extended simulations with simultaneous tuning of initial unmeasured VOC mixing ratios, initial HONO mixing ratios, and $pNO_3^-$ photolysis rates. Results demonstrate that multiple combinations of these processes can reasonably reproduce the age evolution of ozone and some other species. No combination of scaling factors, however, optimizes agreement among all observations.

## 4 Discussion

### 4.1 $NO_y$ Conservation

As noted in Sect 3.1.4, the $\Sigma NO_{y,obs}$ NEMR declines by a factor of 3 over 12 hours of aging. For a well-defined plume, total $NO_y$ should be conserved (Juncosa Calahorrano et al., 2020). The Rim Fire plume is larger and more disperse than most previously-studied wildfires, and SEAC⁴RS observations provide limited information regarding variability in emissions or background concentrations. Potential explanations for the apparent decline of the $\Sigma NO_{y,obs}$ NEMR in the Rim Fire plume include 1) conversion to unmeasured $NO_y$, 2) changing $NO_x$ emission ratios, 3) unmeasured background variability, and 4)

deposition. The last of these is unlikely given that sampling occurred in the uppermost boundary layer and lower free troposphere.

Conversion of measured to unmeasured long-lived $NO_y$ may provide a partial explanation. The SEAC[4]RS measurement suite includes many, but not all, classes of $NO_y$. More recent observations of smaller U.S. wildfires, using new measurement techniques that better speciate organic nitrogen, have shown conservation of the $\Sigma NO_y$ NEMR at physical ages up to 5 h (Juncosa Calahorrano et al., 2020). Comparison to this more recent dataset suggests that unmeasured $NO_y$ (such as complex organic nitrates) might account for $10 - 20\%$ of the $\Sigma NO_{y,obs}$ NEMR decrease in the Rim Fire plume. Photolysis of

unmeasured HONO emissions could buffer this loss by generating $NO_x$ on short timescales (Fig. S17).

        Changing $NO_x$ emission ratios are another possible explanation. As discussed in Sect. 2.3, older samples represent emissions from earlier in the day, when we might expect more smouldering combustion (Wiggins et al., 2020). Other evidence for changing fire phase, including MCE and nitrile NEMRs (Fig. S4), is inconclusive. $NO_x$ emission factors (g per kg fuel burned) can increase with MCE by a factor of 2 or more (Lindaas et al., 2020), and we might expect a similar trend in $NO_x$

ERs. To illustrate potential impacts, we performed sensitivity tests on simulation M1 with initial $NO_x$ multiplied by a factor of 0.5 or 2 (Fig. S24). This nearly spans the range of observed $\Sigma NO_y$. We have relatively more confidence in the $NO_x$ ER at early ages (Liu et al., 2017), so we focus on the half-$NO_x$ case that approaches $\Sigma NO_{y,obs}$ at later ages. Halving initial $NO_x$ reduces model $O_3$ and PAN, increases VOC lifetimes (less OH), and increases peroxide production. Accounting for potential emission changes rigorously in the model would complicate analysis of HONO mechanisms. For example, adding HONO via

initial conditions or $pNO_3^-$ photolysis to increase radical production increases model $NO_y$, necessitating further initial $NO_x$ reduction to maintain agreement with observed $NO_y$. Heterogeneous conversion of $NO_2$ does not alter total $NO_y$, but it also does not amplify ozone. Because of these uncertainties, $\Sigma NO_y$ is a weak constraint on HONO chemistry in this case study. Regional and global model simulations also utilize time-invariant EFs, which likely impacts their representation of diurnal variability of biomass burning chemistry.

Variable background mixing ratios may also impact calculated NEMRs, especially at later ages. For reasons detailed in Sect. 2.1, we assume constant backgrounds. The $\Sigma NO_{y,obs}$ background is 0.7 ppbv. Assuming instead a background of 0 ppbv increases the $\Sigma NO_{y,obs}$ NEMR by a factor of 1.45 at the oldest ages, but this is an extreme lower limit and assumes constant background CO. We have more confidence in the background estimation at early ages, when dilution is strongest. Furthermore, the same background is used for both the modelled and observed NEMRs. Thus, variable backgrounds may have

some influence on observed $NO_y$ NEMRs but a lesser impact on model-measurement comparisons.

        In summary, declining $NO_x$ emission ratios are the most likely explanation for the age dependence of the $\Sigma NO_{y,obs}$ NEMR, but we cannot exclude potential influence from unmeasured $NO_y$ and changing backgrounds. These uncertainties were acknowledged at the outset of the model analysis, and this reinforces the need for caution when interpreting model-measurement agreement at later ages. Nonetheless, the comparison of different simulations can yield insight into the

consequences of augmenting canonical chemistry with new species and reactions.

**4.2 Radical Production and Fate**

Inclusion of additional reactive VOC and HONO significantly accelerates radical throughput. Figures 5a-b summarize rates of key $RO_x$ (OH + $HO_2$ + $RO_2$ + RO) production and loss pathways integrated over the first 2.3 hours of plume aging, where NEMRs change most rapidly. Compared to the base simulation, total $RO_x$ initiation and termination doubles in simulation M1
(VOC addition) and more than triples in M2c (initial HONO). As illustrated in Sect. 3, these changes influence the lifetime of reactive gases and the production of secondary compounds such as ozone, peroxides, organic nitrates, and oVOC. Comparison of $O_3$ NEMRs in Fig. 2k, however, also demonstrates simulation of an ozone profile that reasonably matches observations even with missing processes. Constraints on other aspects of the chemical system facilitate holistic evaluation of additional chemistry.

Photolysis sources dominate radical initiation (Fig. 5a). Photolysis of $O_3$ to $O^1D$ and $H_2O_2$ are each less than 2% of total production in all simulations. In the base simulation, photolysis of HCHO and other oVOC comprises 75% of the radical source. Enhanced reactive VOC in simulation M1 doubles the initiation rate, mainly via further oVOC photolysis and alkene ozonolysis. The largest contributors in the former category are glyoxal (19%) and methyl glyoxal (15%). HONO photolysis in simulations M2 – M4 further accelerates initiation. Simulation M2c (initial HONO) exhibits the most substantial increase of
the plotted simulations, with HONO photolysis comprising 33% of initiation.

        The relative contribution of HONO is smaller here than in other recent studies (Peng et al., 2020; Robinson et al., 2021; Theys et al., 2020) for at least two reasons. First, we average over the first 2.3 hours of aging based on observed rapid ozone production, while other studies may integrate over a shorter timescale when HONO is relatively more important. Second, incorporation of an extended VOC pool greatly enhances oVOC photolysis in our study. oVOC photolysis may be
underestimated in previous radical production estimates: Peng et al. (2020) only account for HCHO and $CH_3CHO$, Theys et al. (2020) account for photolysis of 16 oVOC, and Robinson et al. (2021) does not incorporate VOC beyond those appearing in the MCM or the extended biomass burning mechanism. There is uncertainty in this contribution due to MCM mapping of VOC (Table S2, Fig. S6). Nonetheless, studies failing to account for all oVOC may under-predict radical initiation and over-estimate the relative importance of other radical sources. This may also be the case for urban environments (Qu et al., 2021).

Radical termination includes significant contributions from both $RO_x$-$RO_x$ and $RO_x$-$NO_x$ reactions (Fig. 5b). Formation of peroxides comprises most of the former group, with equal contributions from $HO_2$ + $HO_2$ and $RO_2$ + $HO_2$ in simulation M0. Addition of reactive VOC in M1 doubles the rate of $HO_2$ + $HO_2$ and triples the rate of $RO_2$ + $HO_2$. PN formation comprises 65 – 86% of $RO_x$-$NO_x$ termination, with larger contributions at higher VOC and higher initial HONO. Pernitric acid formation ($HO_2$ + $NO_2$) is 18% of $RO_x$-$NO_x$ termination in simulation S2c, reflecting fast formation in the concentrated plume followed by rapid dilution that outpaces thermal decomposition. Contributions of nitric acid (OH + $NO_2$) and nitroaromatic
(AromNO$_2$) formation are 3 – 15% and 0.1 – 6% of $RO_x$-$NO_x$ termination, respectively. The "OH + X" group includes reaction of OH with HONO and PNs to form $NO_2$ and other products. These reactions remove $RO_x$, but they are not strictly radical termination.

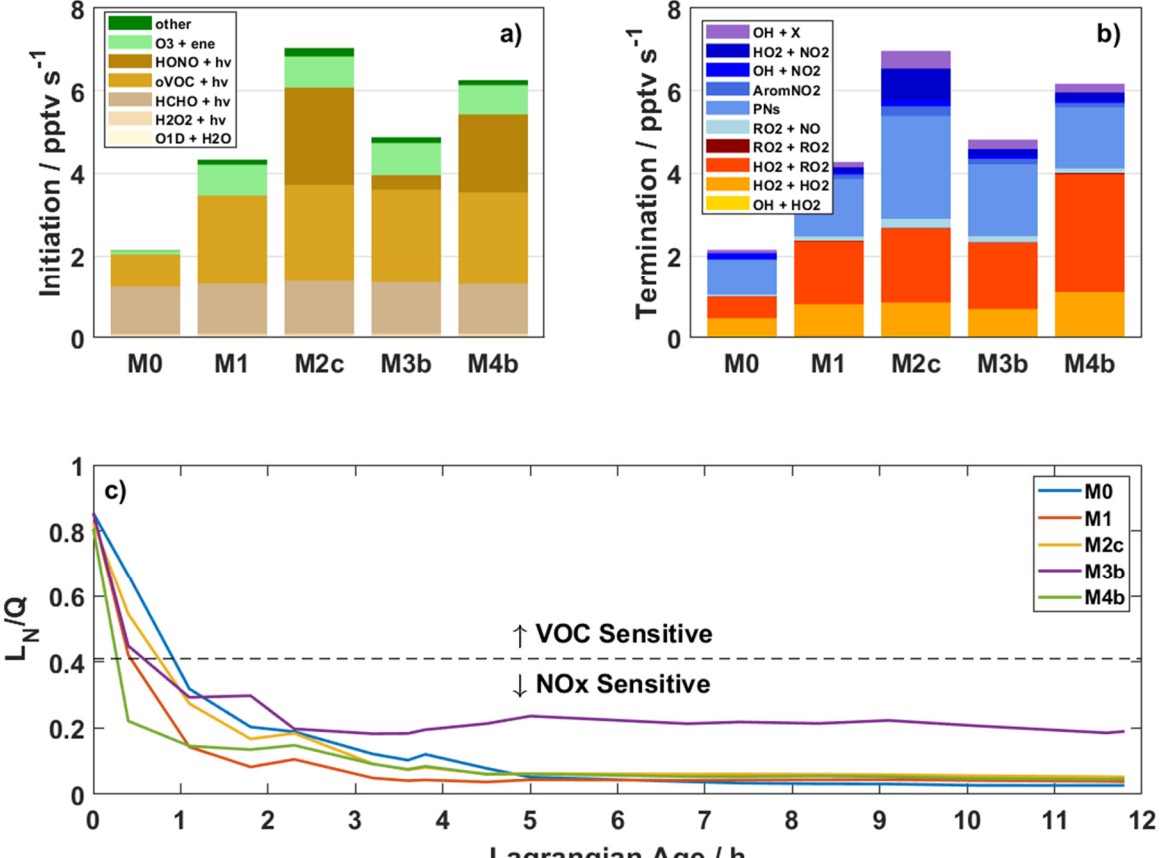

**Figure 5. Upper: rates of RO$_x$ initiation (a) and termination (b) integrated over the first five data points (Lagrangian age 0 – 2.3 hours) via trapezoidal integration. In (a), "oVOC + h$\nu$" includes all oVOC photolysis reactions other than HCHO and "other" includes NO$_3$ + VOC and other minor reactions. In (b), yellow-red and blue shades represent RO$_x$-RO$_x$ and RO$_x$-NO$_x$ reactions, respectively. "AromNO2" represents formation of nitroaromatics and "PNs" is net formation of peroxy nitrates. "OH + X" (purple) represents reactions of OH with organic compounds (typically PNs), which result in formation of NO$_2$ and other non-RO$_x$ products. Lower: Fraction of RO$_x$ radicals lost via reactions with NO$_x$ (c). Model simulations are as described in Fig. 2 and Table 1. The dashed line denotes the approximate transition between NO$_x$-sensitive and VOC-sensitive ozone production for biomass burning chemistry as suggested by Robinson et al. (2021).**

The balance of RO$_x$ loss via NO$_x$ and RO$_x$ sinks, typically represented as the ratio of NO$_x$-related loss (L$_N$) to total

radical loss or production (Q), quantifies the sensitivity of ozone production to VOC and NO$_x$ availability (Kleinman et al., 1997; Kleinman, 2005). Here we define L$_N$ as the sum of losses via formation of ANs, PNs, nitroaromatics, nitric acid, and pernitric acid (blue-shaded reactions in Fig. 5b). Traditionally, the assumption that HNO$_3$ formation is the dominant NO$_x$ sink implies a transition from VOC to NO$_x$-sensitive O$_3$ production at L$_N$/Q = 0.5 (Kleinman et al., 1997). Non-negligible organic nitrate formation alters this threshold (Robinson et al., 2021; Schroeder et al., 2017). We adopt a threshold of L$_N$/Q = 0.41

based on a recent study of Western U.S. wildfires (Robinson et al., 2021).

Ozone production in the young plume is sensitive to both VOC and $NO_x$ (Fig. 5c). All plotted simulations start with a ratio around 0.8 (VOC-sensitive) and transition to $NO_x$-sensitive within the first hour. This rapid shift is consistent with afternoon aging in other Western U.S. wildfires (Robinson et al., 2021) and reflects sequestration of $NO_x$ within PNs and other organic nitrates. Addition of VOC in simulation M1 accelerates this transition (more $RO_x$), while HONO slows the transition in M2c and M3b. Simulation M4b exhibits the fastest transition, where $NO_2$ conversion to HONO amplifies $RO_x$ and mitigates PAN formation.

The consequences of secondary chemistry assumptions become more apparent at later ages. Most simulations decay to steady $L_N/Q$ values of 0.03 – 0.06 after 5 hours of aging. Simulation M3b is the exception, maintaining $L_N/Q$ around 0.2 due to a sustained $NO_x$ source via $pNO_3^-$ photolysis. The accuracy of this simulation is dubious due to a lack of HONO observations; however, the contrast between M3b and M4b illustrates divergent effects of HONO production mechanisms on radical chemistry. Such differences can affect sensitivity to downwind perturbations, such as when mixing with urban (high $NO_x$) or biogenic (high VOC) air.

### 4.3 Implications for Modelling Biomass Burning Chemistry

Box model deficiencies temper the above results and underscore the challenges of simulating smoke plume chemistry. Simulations including both unmeasured reactive VOC and initial HONO or $pNO_3^-$ photolysis (M2c or M3b) reasonably match observed NEMRs in the first 2 hours of aging (Fig. 2); however, none of the scenarios explored here simultaneously reconcile $O_3$, $NO_x$, and PAN throughout the whole sample period. Errors due to assumptions about emissions and air mass history may become more significant further downwind; for example, the model does not capture the large decline in total observed $NO_y$ (Fig. S17). Better observations of emission and background variability would reduce uncertainty, although this can be logistically challenging for large smoke plumes like the Rim Fire.

Errors in kinetics or VOC speciation may also contribute to model uncertainties. For example, our mechanism includes updated reaction rate coefficients for PA + $HO_2$ (faster by a factor of 1.33) and PAA + OH (slower by a factor of 123) based on recent laboratory results (Berasategui et al., 2020; Jenkin et al., 2019). Using MCM default values instead reduces model PAA NEMRs by half (results not shown). Analogous changes to other aspects of the mechanism, due to yet-unrecognized systematic errors, could influence other species. Over-prediction of PAN in simulations M2c and M3b may reflect errors in the thermal equilibrium, the $NO/NO_2$ ratio (under-predicted at later ages in simulation M3b), and/or VOC speciation. Regarding the last issue, acetaldehyde oxidation comprises half of PA production in our simulations, while the other half stems from mostly unmeasured precursors like methyl glyoxal (24% of production) (Fig. S25). Observations of major PAN precursors are necessary to close the PAN budget. Observations of PNs with specific precursors also afford complementary information; for example, over-prediction of APAN in most scenarios (Fig. S14) may imply that initial acrolein is too high. Observations of HONO, $HO_2NO_2$, and total $NO_y$ would also help to fully constrain radical sources, cycling, and fate.

For regional and global models, whether and how to account for the full VOC distribution remains an open challenge. Advances in instrumentation have facilitated quantification of myriad reactive gases (Heald and Kroll, 2020). Cumulatively,

individually minor species can comprise a significant fraction of OH reactivity (Fig. 4) and potential organic aerosol mass (Gilman et al., 2015). Condensed mechanisms cannot represent this level of speciation. One option to reduce this complexity, similar to our methodology, is to identify proxy or surrogate species within a given mechanism based on mass, molecular formula, reactivity, volatility, or other metrics. Given the complexity of smoke plume chemistry, machine learning techniques may also prove useful for condensing VOC into a manageable framework (Kelp et al., 2020). Multifaceted observations of emissions, oxidation products, and reaction intermediates can constrain representation of key chemical processes that influence the spatial and temporal extent of air quality and climate impacts.

Many atmospheric chemistry models do not incorporate primary emissions or heterogeneous production of HONO in standard simulations. Primary HONO is a key oxidant source in young fire plumes (Peng et al., 2020). The HONO photolysis lifetime of 10 – 20 minutes can be on the order of, or much shorter than, an advection timescale for a single grid cell (e.g., a 12 km grid cell at a wind speed of 10 m s$^{-1}$ has an advection timescale of 20 minutes). Thus, "instantaneous dilution" may be a major hurdle to simulating this fast chemistry. Plume heterogeneity (e.g., a darker core) also confounds efforts to simulate chemistry in an average sense (Wang et al., 2021; Palm et al., 2021; Decker et al., 2021b). Putative secondary HONO sources are manifold (Zhang et al., 2019), as are associated uncertainties. A recent lab study of pNO$_3^-$ photolysis on inorganic aerosol suggests a ten-times slower photolysis rate than assumed in simulation M3b (Shi et al., 2021), and a similar reduction was derived from an analysis of NO$_y$ partitioning in polluted marine air (Romer et al., 2018). Other lab work has shown that the rate and product yield of this reaction is sensitive to aerosol surface composition (Ma et al., 2021). Robust parameterizations will require continued systematic study of controlling factors, preferably under environmentally relevant and near-ambient conditions.

Wildfire emissions may have a more sustained influence on regional background chemistry than is currently appreciated. Synergistic increases in OH reactivity and oxidants in our sensitivity simulations stimulate production of multi-generation oVOC and organic NO$_x$ reservoirs like PAN, which may alter the spatiotemporal scale of ozone and organic aerosol production. The modest decline of normalized excess OH reactivity (Fig. 4b) further implies that active chemistry persists far downwind. For example, the 62% decrease over 11.8 h of aging in simulation M3b implies an effective pseudo-first-order photochemical lifetime of 12.2 h for the total OH sink. This aligns with a recent analysis of satellite observations suggesting an effective lifetime of 20 hours or more for formaldehyde and glyoxal in aging smoke plumes (Alvarado et al., 2020). Space-based remote sensing of atmospheric composition may aid evaluation of model biomass burning emissions and chemistry on these scales.

## 5 Conclusions

Using a 0-D puff model constrained with SEAC$^4$RS in situ observations, we have examined the gas-phase chemical evolution of the 2013 California Rim Fire plume and illustrated the sensitivity of chemistry to unmeasured reactive VOC and HONO. The rich measurement suite permits a holistic evaluation of the various components of the chemical system, including VOC,

HO$_x$, NO$_y$, and O$_3$. Initializing with observed gases only, the model reasonably reproduces the evolution of O$_3$ over 12 hours of aging but fails in other key aspects, including over-prediction of reactive VOC decay rates and under-prediction of NO$_x$ and HCHO. Accounting for additional VOC identified by recent laboratory studies increases OH reactivity more than twofold throughout the simulation, drawing down model OH and generally degrading performance. Subsequent addition of HONO amplifies radical production and cycling. Addition of initial HONO (assumed to be primary emissions) or secondary production via pNO$_3^-$ photolysis improves predictions of ozone, oVOC, and NO$_x$, while HONO production via NO$_2$ heterogeneous conversion generally degrades model performance. Further "optimization" simulations demonstrate that multiple combinations of enhanced VOC and primary/secondary HONO can minimize model-measurement bias with respect to O$_3$ and NO$_x$, although we cannot reconcile the model with all observations simultaneously. A decline in total observed NO$_y$ with age may be due to unmeasured NO$_y$, changing NO$_x$ emission ratios, or unmeasured background variability; the exact cause for this behavior remains unresolved. Examination of model reaction rates over the first 2.3 hours of aging demonstrates the potentially dominant contribution of oVOC photolysis to radical initiation, the importance of PN formation as a NO$_x$ sink, and the competitive roles of RO$_x$-RO$_x$ and RO$_x$-NO$_x$ reactions in radical termination. Ozone production is sensitive to both VOC and NO$_x$ in the young plume for all sensitivity simulations. Ozone production transitions from VOC-sensitive to NO$_x$-sensitive after ~1 hour of aging. The timing of this transition depends on VOC and HONO. Downwind NO$_x$ sensitivity depends on the nature and efficacy of assumed secondary HONO/NO$_x$ sources.

A primary finding of this study is that reactive VOC and oxidant sources are complementary in the wildfire-influenced atmosphere, and consideration of both is necessary for accurate simulation of near- and far-field impacts on atmospheric composition and air quality. Future efforts must focus on an efficient solution for incorporating this somewhat novel chemistry in regional and global models, or else quantify the uncertainty associated with neglecting it.

This work also demonstrates the value of a near-comprehensive payload afforded by a heavy-lift aircraft like the NASA DC-8. The breadth of information in the SEAC$^4$RS dataset enables a holistic examination of individual aspects of the chemical system within the context of the whole. Ongoing analysis of data from missions focused on biomass burning will illuminate the factors driving variability in reactive VOC, HONO, and other key aspects of fire plume chemistry. This same approach is beneficial to investigation of other complex environments (urban, biogenic, etc.) that comprise the lower troposphere.

**Appendix A. Photolysis Parameterization**

For each puff, photolysis frequency $J$ at solar time $t$ (and a corresponding Lagrangian plume age) is calculated as

$$J(t, age) = sJ_{cs}(t)/r(age) \tag{A1}$$

Here, $J_{cs}(t)$ is the clear-sky photolysis frequency and $s$ and $r(age)$ are observation-based scaling factors. Clear-sky photolysis frequencies stem from F0AM's hybrid parameterization, which combines solar spectra from the tropospheric ultraviolet and visible radiative transfer model (TUVv5.2, available at https://www2.acom.ucar.edu/modeling/tropospheric-ultraviolet-and-

) with literature-recommended cross sections and quantum yields. This calculation uses actual SZA and measurement altitude. We assume an overhead ozone column of 290 DU and a surface albedo of 0.01. The age-dependent

scaling factor *r(age)* is determined by fitting the ratio of clear-sky to observed photolysis frequencies as a function of Lagrangian age using a modified exponential:

$$r(age) = r_0 e^{-age/\tau} + r_\infty \left(1 - e^{-age/\tau}\right) \tag{A2}$$

Here, $r_0$, $r_\infty$, and $\tau$ are fitting coefficients that vary for each J. Figure A1 illustrates this fit for ozone and $NO_2$ photolysis frequencies, which are suppressed by factors of 3.2 and 1.9, respectively, in the fresh plume. Fits capture $83 - 92\%$ of the

690 variability in the clear-sky to observed ratio for all J-values, with the exception of two points with ages of $1 - 2$ hours. These points are not included in the fit as they appear to be outliers, especially for $J(O^1D)$, and their inclusion would significantly degrade overall fit quality. It is unclear why these points differ from the overall trend, as no other dilution or chemical markers show exceptional behavior at these times. The age correction is the same for all puffs. The other parameter, *s*, is a scalar multiplicative factor that adjusts model J-values to agree with observations at trajectory endpoints (analogous to the scaling

applied for trajectory meteorology) and is different for each puff.

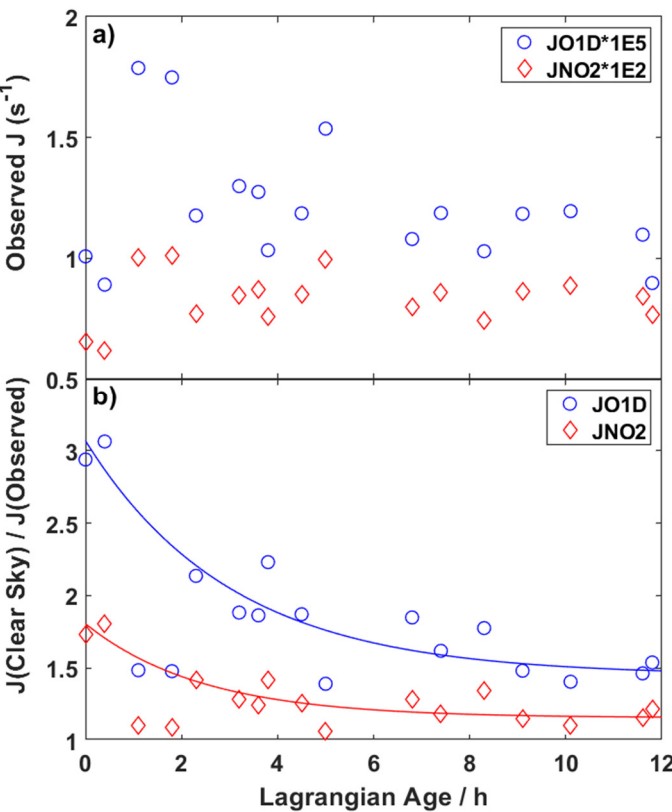

**Figure A1. (a) Average observed photolysis frequencies for each WAS sample for $O_3 \rightarrow O(^1D) + O_2$ (blue circles) and $NO_2 \rightarrow NO + O(^3P)$ (red diamonds). (b) Ratio of clear-sky to observed photolysis frequencies. Clear-sky values are calculated using TUV solar spectra and F0AM parameterizations. Lines represent least-squares fits to Eq. (A2). The two points between ages of 1 – 2 hours are excluded from the fit.**

## Data and Code Availability

Data used in this study is archived at http://doi.org/10.5067/Aircraft/SEAC4RS/Aerosol-TraceGas-Cloud. The F0AM box model is available at https://github.com/AirChem/F0AM and http://doi.org/10.5281/zenodo.5752566. Model setup code is available from the contact author upon request.

## Author Contributions

GMW conceptualized the study, conducted the modelling and analysis, and wrote the manuscript. All authors contributed to data curation and review of the manuscript.

## Competing interests

Some authors are members of the editorial board of ACP. An independent editor guided the peer review process, and the authors have no other competing interests to declare.

## Acknowledgements

The SEAC[4]RS mission was supported by the NASA Tropospheric Composition program and grants from the NASA ROSES SEAC[4]RS program (NNH10ZDA001N, NNX12AC03G, and NNX12AB82G). We thank the DC-8 pilots, crew, payload operators and mission scientists for their hard work and dedication. We thank Luke Ziemba, Lee Thornhill, and the LARGE team for LAS data. We thank Anthony Bucholtz for BBR data. We are also grateful to NASA ESPO for mission logistics. Analysis and modelling were supported by NOAA Climate Program Office's Atmospheric Chemistry, Carbon Cycle, and Climate program (NA17OAR4310004). The Jimenez group acknowledges support from NASA grants 80NSSC19k0124 and 80NSSC18K0630. PTR-MS measurements during SEAC[4]RS were supported by the Austrian Federal Ministry for Transport, Innovation and Technology (bmvit) through the Austrian Space Applications Programme (ASAP) of the Austrian Research Promotion Agency (FFG). A.W. and T.M. received support from the Visiting Scientist Program at the National Institute of Aerospace (NIA). We thank many colleagues for their assistance, insight and feedback, including Steve Brown, Christine Wiedinmyer, Sarah Strode, Ann Marie Carlton, Matt Coggon, Jim Roberts, Joel Thornton, and Qiaoyun Peng.

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
