# Peer review of "Photochemical Evolution of the 2013 California Rim Fire: Synergistic Impacts of Reactive Hydrocarbons and Enhanced Oxidants"

_Atmospheric Chemistry and Physics, 2021_

## Author Response (AR1)

We thank the referees for their thorough and thoughtful comments on the manuscript. In the following, **referee comments are shown in bold**, responses are in plain text, and *altered manuscript text is shown in italics*. All referenced line numbers refer to the original manuscript, not the revision.

**Referee 1**

**The authors use a 0-D puff model to investigate emissions and chemistry in a biomass burning plume from the Rim Fire observed during the SEAC4RS campaign. The time evolution of normalized excess mixing ratios (NEMRs) is constrained by observations of $O_3$, $NO_x$, a large number of $NO_y$ species, and a large number of VOC which help constrain $RO_x$ chemistry. Six model cases are investigated with a particular focus on HONO, 1) using only observed species, 2) adding unobserved VOC based on lab studies, 3) adding HONO as a primary emission, 4) adding HONO via particular nitrate photolysis, 5) adding HONO via heterogeneous reaction of $NO_2$, and 6) a combination of 2,3, and 4. Implications for the representation of the investigated chemistry in other models are briefly discussed.**

**The work is scientifically sound and valuable as a thorough investigation of a case study and is generally well presented. My general comments are to better contextualize the case study, and to provide additional detail on the expanded VOC reactivity and the lack of $NO_y$ closure and how the latter relates to the HONO additions investigated. I elaborate below.**

**The investigation of the case study could benefit from some contextualization and summarization. While there is extensive literature available on the SEAC⁴RS campaign and this flight, certain details should be made available to the reader in this work, e.g. how did the Rim Fire compare to other fires investigated during SEAC⁴RS?**

We have added the following statement to Sect. 2.1. We cannot comment on how the Rim Fire compares to other fires in terms of emissions or chemistry as that would be a separate research topic, and some papers (already referenced in this section) have looked at multiple aspects of multiple fires.

*SEAC⁴RS also investigated several other wild and agricultural fire plumes (Liu et al., 2016; Toon et al., 2016).*

**A single background period is chosen, how does this background compare with other observations?**

Please see our response below to a similar specific question on L130.

**What was the fuel mix for the Rim Fire?**

As stated in Sect. 2.1, the fuel was a mixed conifer forest. This likely consists of multiple species given the size of the fire. More specific information is not available, to our knowledge.

**In a similar vein a number of changes in background are inferred for a variety of species many of which are likely related, e.g. an increase in biogenic background after 2 hours, these can be challenging to keep track of; I would recommend a timeline or concise summary of such changes as a single reference rather than the current references to a variety of sections above and below.**

We have added some text to the beginning of Sect. 3.1 to summarize major transitions in the data.

*Observations illustrate several general features, including 1) rapid oxidation of primary emissions and production of secondary species in the first 2 h of aging, 2) mixing with biogenic emissions around an age*

*of 2 – 3 h, and 3) a transition in some species associated with a decrease in sampling altitude around an age of 6 h (Fig. S1). The following sections survey age-dependent trends in primary VOC, peroxides, oxygenated VOC, reactive nitrogen, and ozone.*

**Realizing that once constructed the Lagrangian age is the time axis used, I would also encourage caution when referring to observed changes as a function of age which are attributed to fire variability. In some instances, age is used to refer to the evolution or the fire which is a separate time axis from the aging of emissions from a given point in time.**

We have searched the manuscript for instances of the use of the word "age" or "aging", and this is always used with reference to the Lagrangian age. This age is inherently tied to the progression of the fire, since changing emissions may be one driver of variability as an (apparent) function of Lagrangian age. We have identified two instances where this distinction may become confusing (lines 178 and 377), and we have attempted to clarify in the text.

L178: *Actual ERs likely vary among the plume samples. Older samples represent fire emissions from earlier in the day, and we might expect these samples to show an increased signature of smoldering relative to flaming due to typical wildfire diurnal progression driven by wind and temperature.*

L377: *Furthermore, a model simulation using lower initial $NO_x$ for successively older puffs (that is, assuming declining $NO_x$ emissions from smoke emitted earlier in the day) would further degrade model prediction of observed $NO_x$ at later ages.*

**Figure 4c shows the large fraction of OH reactivity which arises from unmeasured species particularly aromatics. What is not clear is what fractions of the secondary species arise from these additional species. While the authors make a compelling case that the amount of "missing" OH reactivity cannot be explained by measurement error, it is not clear whether there are additional bounds on this from the modeling results. Understanding that the secondary VOC also changes due to the change in OH and other oxidants, can the authors offer some estimate of what fraction of the OH reactivity from secondary species is due to the unmeasured VOC?**

Yes, we can compare the secondary component from M0 and M1. We have added the following to Sect 3.2.

*Comparing M0 and M1 suggests that 85% of this "secondary" reactivity (1.8 $s^{-1}$) is due to oxidation of unmeasured VOC.*

**Figure 3 shows that the observed $NO_y$, which is nominally a conserved family as defined here, is reduced markedly with increased Lagrangian age. As the authors discuss it is unlikely that any of the component observations are sufficiently far off to explain the discrepancy and major unmeasured components such as HONO, $HO_2NO_2$, and $CH_3O_2NO_2$ are also unlikely to fill the gap leaving open the possibility of unknown $NO_y$ reservoirs. The HONO sensitivity studies introduce mechanisms converting observed $NO_y$ to unobserved $NO_y$ (although it is quickly returned), which provides a useful reference for flux out of observed $NO_y$. How do these compare with the observed rate of loss?**

Please see our response regarding NOy at the bottom of this document. Given uncertainties in the cause of sum-NOy changes, we believe this would be a weak constraint, at best, on HONO production. Also, as

the referee notes, the conversion of HONO back to observed forms of NOy is extremely fast, so there effectively is no "flux out" via HONO production.

**Peng et al., 2020 observed a rapid decline in the HONO NEMR in the first two hours, while Theys et al., 2020 observed a decrease in the HONO/NO₂ ratio on a similar time scale. These ratios are introduced in Sect. 2.4.2 in this work already. HONO time evolution seems to be broadly consistent with both works, but is the trend in either ratio reproduced in this work?**

The trend in the HONO NEMR from Peng (2020) is generally reproduced. The trend in HONO/NO2 is not reported in Peng, and it is difficult to interpret in Theys (2020) as the results shown are from a single fire and there are many uncertainties regarding interpretation of satellite retrievals and comparison to in situ observations. We have modified the supplementary figure and added some notes regarding this in the Sect. 3.3.

[Figure]

*Figure S17. Age evolution of simulated absolute HONO mixing ratios (a), HONO NEMRs (b), and the ratio of HONO to NO₂ (c). Colored lines are as described in Fig. S9.In (b), the black dashed line shows the fitted line from Fig. 3 of Peng et al. (2020). In (c), the shaded gray area is the range of values reported by Peng et al. (2020) and Theys et al. (2020).*

L451: *Initial HONO NEMRs and HONO/NO₂ ratios fall within the general range observed in other fire plumes* (Peng et al., 2020; Theys et al., 2020).

L465: *The HONO NEMR at ages beyond 2 h also aligns with values observed in other wildfires (Fig. S17b).*

L477: *This is also evident in the HONO/NO₂ ratio, which ranges from 0.2 to 3.9 and exceeds the ratio for simulation M3b by a factor of 4 or more (Fig. S17c).*

**For the pNO₃ photolysis case, without a process to covert NOy back to pNO₃ and with pNO₃ constrained to observations is this an unbounded production of gas-phase NOy? Given the dominance of PN as a fraction of observed NOy, is this the principal reason for the reproduction or is HONO chemically particularly well suited to accomplish this? As noted above, the text seems to indicate that there is evidence for unknown NOy reservoirs, is the failure of closure across different criteria in Sect. 3.4 indicative of that or unrelated?**

The production of HONO and NO2 from pNO3 photolysis is not completely unbounded. The rate depends on pNO3, which generally decreases as a puff ages due to dilution. It is also not unreasonable to assume some chemical loss of pNO3 given the decline in pNO3 NEMRs, although the drivers of this trend remain unclear (see our discussion on NOy at the bottom of this document). A fit to the pNO3 NEMR gives a pseudo-first-order e-folding timescale of 9.6 h. For comparison, the (instantaneous at the

time of measurement) lifetime of pNO3- against photolysis in M3b is 3.4 +/- 0.6h. We have added a paragraph regarding this point in Sect. 3.3.

*The supply of pNO$_3^-$ imposes a practical limit on the rate of HONO and NO$_2$ production via this process. Particulate nitrate production and loss is not rigorously modelled in our simulations, but we can estimate the magnitude of this limitation. A linear fit to the observed pNO$_3^-$ NEMR yields an e-folding timescale of 9.6 h, whereas the effective lifetime of pNO$_3^-$ with respect to photolysis in simulation M3b is 3.4 ± 0.6 h (mean and standard deviation, averaged over the endpoints of all puffs). Nitric acid and organic nitrate partitioning to particles may resupply some pNO$_3^-$ downwind; observations in other wildfire plumes show pNO$_3^-$ NEMRs increasing with age* (Juncosa Calahorrano et al., 2020). *Given uncertainties regarding the fate of NO$_y$ (Sect. 4.1), the observed pNO$_3^-$ lifetime is not a strong constraint on the potential chemical loss; however, pNO$_3^-$ photolysis rates are unlikely to be larger than those used in our study, and the comparison with observed gas-phase NO$_y$ (Fig. S17) suggests they may be substantially slower.*

**Technical comments:**

**Line 127: Can the authors provide details on the optimized lag-correlation. What was the method, what was the cost function?**

We have changed this to "maximized cross-correlation." This is a standard method in time series analysis, where we lag one variable relative to the other and maximize the correlation coefficient.

**Line 130: Why a single WAS sample for background? What statistics does this provide? How does this background compare with other measurements from SEAC4RS?**

The nature of the flight pattern and intermittent WAS sampling makes it challenging to constrain background variability. We considered several other samples for background determination, shown by the colored circles in the map below. We determined that these samples contained more fire influence than our background sample (based on HCN and CH3CN); these samples all also had higher O3. And as noted in the text, increased background O3 results in a much lower or even negative O3 NEMR.

[Figure]

We have added some text to this section to further clarify this point.

*Background mixing ratios are averaged over a single WAS sample collected east of the plume (orange star in Fig. 1). We also explored using observations upwind, downwind, or west of the plume for background estimation, but these samples contain stronger fire influence than Eastern sample based on the conserved fire tracers HCN and $CH_3CN$. These alternative background samples also contain higher $O_3$ (60 – 80 ppbv vs 50 ppbv), leading to smaller or negative $O_3$ NEMRs and significantly poorer agreement with the model. The influence of background selection on NEMRs depends on the relative magnitude of background and in-plume mixing ratios, which varies with chemical species and age. While this introduces additional uncertainty in observed NEMRs, use of the same backgrounds in model simulations reduces the impact of such uncertainties with regard to model-measurement comparisons.*

**Line 146: It would be helpful for reproducibility to know which algorithm was used to compute the geometric median, and to what precision.**

We have added a reference to the utilized function. The precision is not critical for our study. The averaged trajectories are reasonable compared to the individual trajectories (not shown) and the MODIS imagery of the smoke plume.

Zhong, D.: medoid and geometric median, MATLAB Central File Exchange, https://www.mathworks.com/matlabcentral/fileexchange/70145-medoid-and-geometric-median, 2021.

**Line 149: I assume the authors mean here when the back trajectory first intersected the fire which would be when the trajectory last intersected the fire. I suggest rewording for clarity.**

Corrected.

**Line 152: When was the wind measurement for the transit time estimate taken? At observation or at time of emission? Does the 1h transit time generally comport with the back trajectories?**

The calculation is based on observed wind speed in the near-source sample, which is 11.3 m/s. For comparison, median trajectories in this area have average wind speeds of 11.4 +/- 1.6 m/s.  We have clarified this in the text.

*Based on observed wind speed and modelled trajectories near the fire, we estimate a transit time of ~1 hour for emissions from the Southern-most fire front to reach the North end of the box.*

**Line 178: "age-dependent" here should be substituted with "time-dependent" or something similar. If I understand correctly I would reserve "age" for the evolution of the trajectories and not to refer to different times of emission to avoid confusion.**

We have changed this to time-dependent.

**Line 204: It is written "Heterogeneous chemistry is explicitly included." I assume there is a "not" missing as this is a paragraph on limitations of the model.**

Corrected.

**Line 222: Why report the fuel composition if it is not assessed? Can the authors provide any information relating to this?**

We have removed the sentence reporting % fuel contributions.

**Line 235: Can the authors provide some assessment of the bulk characteristics converting compounds using this method? e.g. total carbon, average molecular elemental composition.**

We have added the following text and supplementary figure.

*Figure S6 compares bulk chemical metrics between the non-MCM species of Koss et al. (2018) and assigned MCM proxies. Overall, proxies reproduce the distribution of OH reactivity (total 47 $s^{-1}$) and carbon content (390 ppbvC) and are biased high with respect to oxygen content and molecular weight.*

[Figure]

*Figure S6. Comparison of chemical metrics for the non-MCM unmeasured VOC from Koss et al. (2018) and their MCM proxies (see Sect. 2.4.1 and Table S2). (a) Molecular weight, (b) number of carbons per molecule, (c) number of oxygens per molecule, (d) oxygen/carbon ratio per molecule, (e) OH reaction rate coefficient, and (f) initial OH reactivity. Note that (e) and (f) are on a log scale.*

**Line 252: How is the $pNO_3^-$ at the start of the puff constrained? If I understand correctly this is nominally the fire, which observations are taken to correspond to this?**

As with gas concentration, initial pNO3 is taken from the first near-source sample. We have added this detail to the text.

**Line 266: The linear relation is valid for the period of the observations, but $J_{NO2}$ should not be linear with SZA as a general rule especially near twilight. The value of $J_{NO2}$ at sunrise is negative using the equation. Is this relation extrapolated back in the puff model to the time of emission, if so what is the nominal SZA at emission?**

There may be some confusion here involving this parameterization. $J_{NO2}$ is derived from the model's photolysis parameterization, not a linear relationship. The linear relationship used here is between total solar irradiance and SZA, which is shown in Fig. S9. It is true that the relationship likely deviates at high

SZA; extrapolation gives Rad = 0 at SZA = 84 degrees, and negative Rad at higher SZA. However, this is not a major issue for us as gamma(NO2) is independent of Rad at values lower than 400 W m$^{-2}$.

We have added a note in the text that JNO2 is from the model, and we have added the following statement regarding the irradiance estimation.

*Some minor systematic bias in Rad may result from extrapolation of this relationship.*

**Line 268: Should the values <10$^{-6}$ not be substituted by 1×10$^{-6}$ based on eq. 4? When multiplying the rate by 1000, is this lower limit similarly scaled?**

Yes. There may be some confusion here regarding the stated ranges for gamma and ka. We have added some text to clarify.

*Figures S6c-d show calculated $\gamma$ and $k_a$ at the end point of each puff. The uptake coefficient ranges from $4 - 11 \times 10^{-6}$, while $k_a$ ranges from $0.3 - 4.9 \times 10^{-6}$ s$^{-1}$. Note that $k_a$ varies along the trajectory for each puff due to changes in radiation and $S_a$. Simulation M4a uses this default parameterization, while in simulation M4b $k_a$ is multiplied by a factor of 1000.*

**Line 344: CH$_3$CHO is not subscripted here.**

Corrected.

**Line 395: I would move Text S2 to main text above to support this statement regarding oVOC production.**

The discussion regarding oVOC is in the SI because most of these compounds do not vary much, and the trends of MVK + MACR tell the same story as isoprene and HCHO, which are shown in the main text. We have modified this sentence to be more specific.

*Production of peroxides and HCHO is too slow, indicating missing sources of peroxy radicals and organic carbon.*

**Line 402-03: I do not understand what the sentence "After 12 h, 32% of M1-simulated OH reactivity is comprised of nearly 2200 species, mostly oxygenated VOC." Is seeking to communicate.**

We have changed this sentence to clarify.

*After 12 h, 32% of M1-simulated OH reactivity is comprised of over 2100 species that are, individually, not very abundant (Fig. 4c, gray area).*

**Line 496: This estimate is also substantially smaller than that in Theys et al., 2020 do the same reasons apply?**

Potentially. The Theys study states that they only use 16 VOC in their radical calculation, and Peng only used 2 (HCHO and CH3CHO). It is also not clear how Theys arrived at their "2/3 of OH production in fresh wildfire plumes" number, e.g. over what spatial/temporal scale the averaging was done. "Fresh" is ambiguous. We have chosen to define our integration scale over the period of most active ozone production.

We have added some further discussion to this section.

*The relative contribution of HONO is smaller here than in other recent studies (Peng et al., 2020; Robinson et al., 2021; Theys et al., 2020) for at least two reasons. First, we average over the first 2.3 hours of aging based on observed rapid ozone production, while other studies may integrate over a shorter timescale when HONO is relatively more important. Second, incorporation of an extended VOC pool greatly enhances oVOC photolysis in our study. oVOC photolysis may be underestimated in previous radical production estimates: Peng et al. (2020) only account for HCHO and CH$_3$CHO, Theys et al. (2020) account for photolysis of 16 oVOC, and Robinson et al. (2021) does not incorporate VOC beyond those appearing in the MCM or the extended biomass burning mechanis*m.

**Line 507: Avoid Ar for Aromatic reserve for argon**

We have changed this to AromNO$_2$.

**Referee 2**

**Wolfe et al use observations and a photochemical model to examine the complex photochemistry in one 2013 fire. The results show that unmeasured VOCs and OVOCs have a huge impact on the photochemistry, which is an important and believable conclusion. However there are several significant uncertainties. The major uncertainty that is discussed is the sources of HONO, which ends up as a downwind NOx source. A major uncertainty that is less discussed is the fate and measurements of NOy species. In particular Figure 3 shows a substantial decline in NOy/CO, implying a loss of NOy other than dilution. Where does this NOy go? Is it lost to deposition? Deposition seems unlikely in this timescale and given particle size distribution. Or is it transformed to other species that are not being measured? How would this impact your conclusions?**

We have added a significant amount of text and figures regarding NOy. Please see the bottom of this document.

**Finally I suggest a bit more on the key measured species. I see in the SI the list of measurements, but I recommend that a bit more be added to the main manuscript to clarify key points (like how NOy was obtained, whether HONO was measured or not…)**

We have added a paragraph to Sect. 2.1 detailing the measurements.

*Table S1 lists the instruments and measurement accuracy for observations used in this study; the payload is further described in Toon et al. (2016). Here we provide a brief summary of key measurements. Most speciated VOC observations (alkanes, alkenes, aromatics, terpenes, and alkyl nitrates) derive from the Whole Air Sampler (WAS), with a sample collection time of 40 s and a sampling interval of 2 – 10 min. We also use VOC and oxygenated VOC (oVOC) observations from the Proton Transfer quadrupole Mass Spectrometer (PTR-MS), including acetaldehyde, the sum of methyl vinyl ketone and methacrolein (MVK + MACR), and the sum of isoprene and furan. Furan is calculated as the difference between the PTR-MS sum and WAS isoprene. Formaldehyde (HCHO) is measured via both laser-induced fluorescence and infrared absorption spectroscopy. Peroxides, nitric acid, and hydroxynitrates are measured via $CF_3O^-$ chemical ionization mass spectrometry (CIMS). Other $NO_y$ measurements include NO and $NO_2$ via chemiluminescence; $NO_2$, total peroxy nitrates, and total alkyl nitrates via thermal dissociation – laser-induced fluorescence; and speciated peroxyacyl nitrates via thermal dissociation iodide CIMS. Ozone is measured via chemiluminscence. Carbon monoxide (CO) is measured via differential absorption. Photolysis frequencies derive from observed up- and down-welling actinic flux combined with literature-recommended cross sections and quantum yields. Other observations used primarily for model inputs include pressure, temperature, water vapour (open-path absorption), particulate nitrate (aerosol mass spectrometer), aerosol surface area (laser aerosol spectrometer), and total solar irradiance. Aside from WAS data, observations are nominally reported at 1 Hz but may contain gaps due to normal instrument operation. For the present study, all 1 Hz data are averaged to WAS collection windows. The dataset does not include observations of total $NO_y$, nitrous acid (HONO), or ammonia ($NH_3$).*

**On the presentation, most aspects are done fairly well with the exception of Figure 5. I found this figure very difficult to interpret. I think the authors are trying to cram too much into one figure and the result is a figure that is very difficult (impossible) to interpret.**

We agree. This is a complicated model experiment to analyze simply, and it does not add much to the paper. We have moved this section to SI Text S4 and added a brief mention about the key result in the end of Sect. 3.3:

*Additional reactive VOC and HONO chemistry collectively improve model-measurement agreement for most observed species, but the implementation of each process is not quantitatively independent. Model performance relative to observations inherently requires a balance between oxidant sinks (VOC) and sources (HONO), both of which are uncertain. SI Text S4 describes extended simulations with simultaneous tuning of initial unmeasured VOC mixing ratios, initial HONO mixing ratios, and $pNO_3^-$ photolysis rates. Results demonstrate that multiple combinations of these processes can reasonably reproduce the age evolution of ozone and some other species. No combination of scaling factors, however, optimizes agreement among all observations.*

**Detailed comments:**

**Line 65: Grammar.**

Fixed, possibly. We do not see any obvious grammar errors here.

**Line 170: Baylon 2018 has a good discussion of the UV impacts on JNO2 and JO3 (https://doi.org/10.1002/2017JD027341)**

We have added a reference to this paper.

**Line 179: Why would MCE decline with age?**

At older ages, emissions occurred earlier in the day, and the general progression of a fire is to go transition from flaming to smouldering later in the day due to meteorology. We have added some text to clarify this:

*Actual ERs likely vary among the plume samples. Older samples represent fire emissions from earlier in the day, and we might expect these samples to show an increased signature of smouldering relative to flaming due to typical wildfire diurnal progression (Wiggins et al., 2020). Observations, however, do not conclusively indicate a time-dependent trend in ERs. MCE generally declines with age (Fig. S4a), consistent a shift from smouldering to flaming over time, but frequent deviation from the expected wildfire value range of 0.8 – 1.0 (Akagi et al., 2011) suggests non-emission influence (e.g., background $CO_2$ variability) that degrades this metric as a combustion phase tracer at later ages.*

**Line 180: Assumption of constant NOy EFs seems important. Evidence?**

We are assuming constant ERs (emission ratios to CO), not constant EFs (emission factors, which are relative to the amount of biomass burned). The arguments for this assumption are presented in this section. We cannot constrain ER variability over time with the available data, and the available evidence is ambiguous, so we assume constant ERs. Please also see our discussion regarding NOy conservation at the bottom of this document.

**Line 204: What het chemistry? Does this include HOx loss on aerosols? Contradicts line 254.**

This was a typo and has been corrected. The model does not have heterogeneous chemistry.

**Line 205:  Measurement accuracy…?  Don't understand this sentence?  I think variability is more important…**

We do not have constraints on variability for the WAS data, which is inherently averaged over $30 - 40$ seconds. It would be possible to calculate standard deviation for fast data over the WAS averaging windows, but it is not obvious that this is a more appropriate metric when evaluating model performance against observations. Indeed, some species might exhibit small variability even though the observations are highly uncertain due to calibration limitations (e.g., PAA and H2O2). Thus, we believe that accuracy is the appropriate metric for the error bars.

We have added a clause to specify that we are drawing a contrast between accuracy and precision, as the latter can be more important at low concentrations.

**Line 227: Change to "reduced by factors of 2.3 and 10…"**

Fixed.

**Line 239:  This discussion on HONO emissions/confusing is confusing.  Wasn't HONO measured, so why do you need these scaling factors?**

No, HONO was not measured. We have clarified this in Sect. 2.1 and reiterated it here.

**Line 244:  Don't understand P-HNO3 photolysis scaling…**

We have reworded this sentence.

*The photolysis frequency for reaction R1 is calculated as $286*J(HNO_3)$ following Ye et al. (2017, 2018).*

**Line 296: Change to "VOC-to-CO emission ratio increase…"**

Fixed.

**Line 313: Fig. S12 is discussed before Fig. S11.**

We have corrected the order of all SI figures.

**Fig 3 and discussion…. Does NOy include p-NO3?   Why does NOy/CO decline?   Where does NOy go?   This seems like a sig uncertainty in the results.   How does this impact the results?**

We have added a significant amount of text and figures regarding NOy. Please see the bottom of this document.

**Line 376:   modeled NOy-obs.   So what does M0 tell us?   It seems the bulk of NOy loss is in the first hour and is likely due to chemistry… So doesn't this imply that the NOy is probably going into unmeasured species?**

We have added a significant amount of text and figures regarding NOy. Please see the bottom of this document. NOy loss is persistent throughout the observational period.

**Lines 391–392: Why does the base simulation over-predict the NO/NO$_2$ ratio during the first few hours?**

We have added the following sentence.

*Disagreement at young ages is consistent with insufficient conversion of NO to NO₂, possibly due to insufficient ozone and/or peroxy radicals.*

**Line 452: The figure for ozone is Fig. 2k, rather than Fig. 2l.**

Fixed

**Lines 457–458: The sensitivity simulations to heterogeneous reaction of NO₂ are in Fig. S20, not Fig. S19.**

Corrected.

**Lines 517–522: Since Table S1 indicates that HCHO and NO₂ measurements were made during the SEAC⁴RS campaign, it might be useful to examine the trend in the HCHO/NO₂ ratio to see if that is consistent with your conclusions regarding the O₃ production regime based on L_N/Q.**

The ratio of HCHO/NO2 is not indicative of ozone production regime in the first few hours due to strong HCHO emissions, and this is when the transition occurs and ozone production is most active. The plot below shows this ratio for the observations and model. Even at older ages, the ratio is much larger than what is typically found in urban environments for the ratio of total columns (2 − 4). Also ozone production is minimal at these ages, and the difference between model runs is mostly driven by variability in NO2. Thus, we do not think this is a useful metric for our study case.

[Figure]

**Line 614: Grammar.**

We have broken this into 2 sentences.

**SI:**

**Text S2: Fig. S12 – not Fig. S13 – shows the age progression of other oVOCs.**

Fixed.

**Text S3: Fig. S13 – not Fig. S14 – shows the results for the other speciated PNs.  In addition, Fig. S14 – not Fig. S15 – shows the results for ΣPN and ΣAN.**

Fixed.

**Fig. S9: The figure caption does not state what the green lines show.**

Fixed.

**Fig. S10 caption: In the first sentence, NMB should be enclosed in parentheses.  In the second sentence, the reference should be cited as Gustafson and Yu (2012).**

Fixed.

**Fig. S16: In the legend, the blue and red solid lines denote the NO$_x$ NEMRs for simulations M0 and M1, respectively, rather than S0 and S1.**

Fixed.

**Figs. S18–S20: The figure captions say Figure 18, 19, and 20, instead of S18, S19, and S20.**

Fixed.

**Fig. S19: It appears that the green lines in panels (j) and (k) are not entirely visible.  If this is true, then the y-axis limit needs to be increased accordingly.**

Fixed.

Regarding NOy Conservation

Multiple comments from both Referees highlight uncertainties regarding the decrease in observed NOy. We did not want this to be central to the study, as we have little observational evidence to discern the reasons for this trend, and the best we can do is speculate. We have added some text to Sect 3.1.4, a new Section to the discussion (Sect. 4.1), and 3 SI figures to confront the uncertainties inherent in this result.

Sect 3.1.4: *Underlying this trend is a rapid NO$_x$ decay, a step-change in pNO$_3^-$ at an age of 2 h, and a gradual decline in ΣPNs (Fig. SX). Figure SXX compares observed gas-phase NOy (ΣNO$_{y,gas}$, which excludes pNO$_3^-$ as this is not modelled) with the model-equivalent sum. The observed gas-phase sum decreases over age from 12.5 to 4.4 ppbv ppmv$^{-1}$ and is ~7 ppbv ppmv$^{-1}$ throughout simulation M0. The causes and consequences of this discrepancy are further discussed in Sect. 4.1.*

[Figure]

[Figure]

| Figure SX. Comparison of age trends for different components of observed NO$_y$, including NO$_x$ (blue circles), total peroxy nitrates (red squares), total alkyl nitrates (yellow triangles), nitric acid (purple X), and particulate nitrate (green stars). | Figure SXX. Age evolution of modelled gas-phase NO$_y$. Symbols and lines are as described Fig. S9. Observed values represent the sum of NO$_x$, ΣPN, ΣHN, and HNO$_3$. Model values represent the sum over the same modelled species and thus exclude HONO, HO$_2$NO$_2$, and nitroaromatics. |
|---|---|

*4.1 NO$_y$ Conservation*

*As noted in Sect 3.1.4, the ΣNO$_{y,obs}$ NEMR declines by a factor of 3 over 12 hours of aging. For a well-defined plume, total NO$_y$ should be conserved (Juncosa Calahorrano et al., 2020). The Rim Fire plume is larger and more disperse than most previously-studied wildfires, and SEAC$^4$RS observations provide limited information regarding variability in emissions or background concentrations. Potential explanations for the apparent decline of the ΣNO$_{y,obs}$ NEMR in the Rim Fire plume include 1) conversion to unmeasured NO$_y$, 2) changing NO$_x$ emission ratios, 3) unmeasured background variability, and 4) deposition. The last of these is unlikely given that sampling occurred in the uppermost boundary layer and lower free troposphere.*

*Conversion of measured to unmeasured long-lived NO$_y$ may provide a partial explanation. The SEAC$^4$RS measurement suite includes many, but not all, classes of NO$_y$. More recent observations of smaller U.S. wildfires, using new measurement techniques that better speciate organic nitrogen, have shown conservation of the ΣNO$_y$ NEMR at physical ages up to 5 h (Juncosa Calahorrano et al., 2020). Comparison to this more recent dataset suggests that unmeasured NO$_y$ (such as complex organic nitrates) might account for 10 – 20% of the ΣNO$_{y,obs}$ NEMR decrease in the Rim Fire plume. Photolysis of unmeasured HONO emissions could buffer this loss by generating NO$_x$ on short timescales (Fig. S17).*

*Changing NO$_x$ emission ratios are another possible explanation. As discussed in Sect. 2.3, older samples represent emissions from earlier in the day, when we might expect more smouldering combustion (Wiggins et al., 2020). Other evidence for changing fire phase, including MCE and nitrile NEMRs (Fig. S4), is inconclusive. NO$_x$ emission factors (g per kg fuel burned) can increase with MCE by a factor of 2 or more (Lindaas et al., 2020), and we might expect a similar trend in NO$_x$ ERs. To illustrate potential impacts, we performed sensitivity tests on simulation M1 with initial NO$_x$ multiplied by a factor of 0.5 or 2 (Fig. S24). This nearly spans the range of observed ΣNO$_y$. We have relatively more confidence in the NO$_x$ ER at early ages (Liu et al., 2017), so we focus on the half-NO$_x$ case that approaches ΣNO$_{y,obs}$*

*at later ages. Halving initial $NO_x$ reduces model $O_3$ and PAN, increases VOC lifetimes (less OH), and increases peroxide production. Accounting for potential emission changes rigorously in the model would complicate analysis of HONO mechanisms. For example, adding HONO via initial conditions or $pNO_3^-$ photolysis to increase radical production increases model $NO_y$, necessitating further initial $NO_x$ reduction to maintain agreement with observed $NO_y$. Heterogeneous conversion of $NO_2$ does not alter total $NO_y$, but it also does not amplify ozone. Because of these uncertainties, $\Sigma NO_y$ is a weak constraint on HONO chemistry in this case study. Regional and global model simulations also utilize time-invariant EFs, which likely impacts their representation of diurnal variability of biomass burning chemistry.*

*Variable background mixing ratios may also impact calculated NEMRs, especially at later ages. For reasons detailed in Sect. 2.1, we assume constant backgrounds. The $\Sigma NO_{y,obs}$ background is 0.7 ppbv. Assuming instead a background of 0 ppbv increases the $\Sigma NO_{y,obs}$ NEMR by a factor of 1.45 at the oldest ages, but this is an extreme lower limit and assumes constant background CO. We have more confidence in the background estimation at early ages, when dilution is strongest. Furthermore, the same background is used for both the modelled and observed NEMRs. Thus, variable backgrounds may have some influence on observed $NO_y$ NEMRs but a lesser impact on model-measurement comparisons.*

*In summary, declining $NO_x$ emission ratios are the most likely explanation for the age dependence of the $\Sigma NO_{y,obs}$ NEMR, but we cannot exclude potential influence from unmeasured $NO_y$ and changing backgrounds. These uncertainties were acknowledged at the outset of the model analysis, and this reinforces the need for caution when interpreting model-measurement agreement at later ages. Nonetheless, the comparison of different simulations can yield insight into the consequences of augmenting canonical chemistry with new species and reactions.*

[Figure]

*Figure SXXX. Age evolution of NEMRs for sensitivity simulations to initial $NO_x$. Simulation M1 (blue line) is modified by multiplying initial NO and $NO_2$ mixing ratios by a factor of 0.5 (red) or 2 (yellow). Other details are as described in Fig. 2 of the main text.*